# Dataset Meta-Learning from Kernel Ridge-Regression

**Timothy Nguyen**    **Zhourong Chen**    **Jaehoon Lee**
Google Research
`{timothycnguyen, zrchen, jaehlee}@google.com`

## Abstract

One of the most fundamental aspects of any machine learning algorithm is the training data used by the algorithm. We introduce the novel concept of $\epsilon$-approximation of datasets, obtaining datasets which are much smaller than or are significant corruptions of the original training data while maintaining similar model performance. We introduce a meta-learning algorithm called Kernel Inducing Points (`KIP`) for obtaining such remarkable datasets, inspired by the recent developments in the correspondence between infinitely-wide neural networks and kernel ridge-regression (KRR). For KRR tasks, we demonstrate that `KIP` can compress datasets by one or two orders of magnitude, significantly improving previous dataset distillation and subset selection methods while obtaining state of the art results for MNIST and CIFAR-10 classification. Furthermore, our `KIP`-learned datasets are transferable to the training of finite-width neural networks even beyond the lazy-training regime, which leads to state of the art results for neural network dataset distillation with potential applications to privacy-preservation.

## 1 Introduction

Datasets are a pivotal component in any machine learning task. Typically, a machine learning problem regards a dataset as given and uses it to train a model according to some specific objective. In this work, we depart from the traditional paradigm by instead optimizing a dataset with respect to a learning objective, from which the resulting dataset can be used in a range of downstream learning tasks.

Our work is directly motivated by several challenges in existing learning methods. Kernel methods or instance-based learning (Vinyals et al., 2016; Snell et al., 2017; Kaya & Bilge, 2019) in general require a support dataset to be deployed at inference time. Achieving good prediction accuracy typically requires having a large support set, which inevitably increases both memory footprint and latency at inference time—*the scalability issue*. It can also raise *privacy concerns* when deploying a support set of original examples, e.g., distributing raw images to user devices. Additional challenges to scalability include, for instance, the desire for rapid hyper-parameter search (Shleifer & Prokop, 2019) and minimizing the resources consumed when replaying data for continual learning (Borsos et al., 2020). A valuable contribution to all these problems would be to find surrogate datasets that can mitigate the challenges which occur for naturally occurring datasets without a significant sacrifice in performance.

This suggests the following

**Question:** *What is the space of datasets, possibly with constraints in regards to size or signal preserved, whose trained models are all (approximately) equivalent to some specific model?*

In attempting to answer this question, in the setting of supervised learning on image data, we discover a rich variety of datasets, diverse in size and human interpretability while also robust to model architectures, which yield high performance or state of the art (SOTA) results when used as training data. We obtain such datasets through the introduction of a novel meta-learning algorithm called Kernel Inducing Points (`KIP`). Figure 1 shows some example images from our learned datasets.

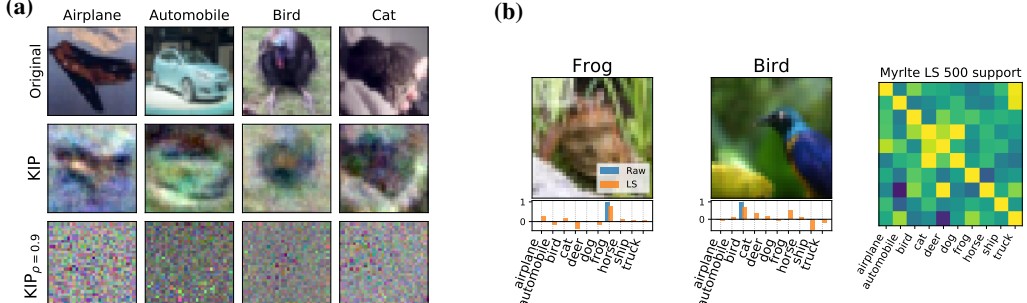

Figure 1: **(a)** Learned samples of CIFAR-10 using KIP and its variant KIP$_\rho$, for which $\rho$ fraction of the pixels are uniform noise. Using 1000 such images to train a 1 hidden layer fully connected network results in 49.2% and 45.0% CIFAR-10 test accuracy, respectively, whereas using 1000 original CIFAR-10 images results in 35.4% test accuracy. **(b)** Example of labels obtained by label solving (LS) (left two) and the covariance matrix between original labels and learned labels (right). Here, 500 labels were distilled from the CIFAR-10 train dataset using the the Myrtle 10-layer convolutional network. A test accuracy of 69.7% is achieved using these labels for kernel ridge-regression.

We explore KIP in the context of compressing and corrupting datasets, validating its effectiveness in the setting of kernel-ridge regression (KRR) and neural network training on benchmark datasets MNIST and CIFAR-10. Our contributions can be summarized as follows:

## 1.1 SUMMARY OF CONTRIBUTIONS

- We formulate a novel concept of $\epsilon$-approximation of a dataset. This provides a theoretical framework for understanding dataset distillation and compression.

- We introduce Kernel Inducing Points (KIP), a meta-learning algorithm for obtaining $\epsilon$-approximation of datasets. We establish convergence in the case of a linear kernel in Theorem 1. We also introduce a variant called Label Solve (LS), which gives a closed-form solution for obtaining distilled datasets differing only via labels.

- We explore the following aspects of $\epsilon$-approximation of datasets:

  1. **Compression (Distillation) for Kernel Ridge-Regression:** For kernel ridge regression, we improve sample efficiency by over one or two orders of magnitude, e.g. using 10 images to outperform hundreds or thousands of images (Tables 1, 2 vs Tables A1, A2). We obtain state of the art results for MNIST and CIFAR-10 classification while using few enough images (10K) to allow for in-memory inference (Tables A3, A4).
  2. **Compression (Distillation) for Neural Networks:** We obtain state of the art dataset distillation results for the training of neural networks, often times even with only a single hidden layer fully-connected network (Tables 1 and 2).
  3. **Privacy:** We obtain datasets with a strong trade-off between corruption and test accuracy, which suggests applications to privacy-preserving dataset creation. In particular, we produce images with up to 90% of their pixels corrupted with limited degradation in performance as measured by test accuracy in the appropriate regimes (Figures 3, A3, and Tables A5-A10) and which simultaneously outperform natural images, in a wide variety of settings.

- We provide an open source implementation of KIP and LS, available in an interactive Colab notebook[1].

## 2 SETUP

In this section we define some key concepts for our methods.

---

[1] https://colab.research.google.com/github/google-research/google-research/blob/master/kip/KIP.ipynb

**Definition 1.** A *dataset* in $\mathbb{R}^d$ is a set of $n$ distinct vectors in $\mathbb{R}^d$ for some $n \geq 1$. We refer to each such vector as a *datapoint*. A dataset is *labeled* if each datapoint is paired with a label vector in $\mathbb{R}^C$, for some fixed $C$. A datapoint along with its corresponding label is a *labeled datapoint*. We use the notation $D = (X, y)$, where $X \in \mathbb{R}^{n \times d}$ and $y \in \mathbb{R}^{n \times C}$, to denote the tuple of unlabeled datapoints $X$ with their corresponding labels $y$.

We henceforth assume all datasets are labeled. Next, we introduce our notions of approximation, both of functions (representing learned algorithms) and of datasets, which are characterized in terms of performance with respect to a loss function rather than closeness with respect to a metric. A loss function $\ell : \mathbb{R}^C \times \mathbb{R}^C \to \mathbb{R}$ is one that is nonnegative and satisfies $\ell(z, z) = 0$ for all $z$.

**Definition 2.** Fix a loss function $\ell$ and let $f, \tilde{f} : \mathbb{R}^d \to \mathbb{R}^C$ be two functions. Let $\epsilon \geq 0$.

1. Given a distribution $\mathcal{P}$ on $\mathbb{R}^d \times \mathbb{R}^C$, we say $f$ and $\tilde{f}$ are *weakly $\epsilon$-close* with respect to $(\ell, \mathcal{P})$ if

$$\left| \mathbb{E}_{(x,y)\sim\mathcal{P}} \Big( \ell(f(x), y) \Big) - \mathbb{E}_{(x,y)\sim\mathcal{P}} \Big( \ell(\tilde{f}(x), y) \Big) \right| \leq \epsilon. \tag{1}$$

2. Given a distribution $\mathcal{P}$ on $\mathbb{R}^d$ we say $f$ and $\tilde{f}$ are *strongly $\epsilon$-close* with respect to $(\ell, \mathcal{P})$ if

$$\mathbb{E}_{x\sim\mathcal{P}} \Big( \ell(f(x), \tilde{f}(x)) \Big) \leq \epsilon. \tag{2}$$

We drop explicit reference to $(\ell, \mathcal{P})$ if their values are understood or immaterial.

Given a learning algorithm $A$ (e.g. gradient descent with respect to the loss function of a neural network), let $A_D$ denote the resulting model obtained after training $A$ on $D$. We regard $A_D$ as a mapping from datapoints to prediction labels.

**Definition 3.** Fix learning algorithms $A$ and $\tilde{A}$. Let $D$ and $\tilde{D}$ be two labeled datasets in $\mathbb{R}^d$ with label space $\mathbb{R}^C$. Let $\epsilon \geq 0$. We say $\tilde{D}$ is a *weak $\epsilon$-approximation* of $D$ with respect to $(\tilde{A}, A, \ell, \mathcal{P})$ if $\tilde{A}_{\tilde{D}}$ and $A_D$ are weakly $\epsilon$-close with respect to $(\ell, \mathcal{P})$, where $\ell$ is a loss function and $\mathcal{P}$ is a distribution on $\mathbb{R}^d \times \mathbb{R}^C$. We define *strong $\epsilon$-approximation* similarly. We drop explicit reference to (some of) the $\tilde{A}, A, \ell, \mathcal{P}$ if their values are understood or immaterial.

We provide some justification for this definition in the Appendix. In this paper, we will measure $\epsilon$-approximation with respect to 0-1 loss for multiway classification (i.e. accuracy). We focus on weak $\epsilon$-approximation, since in most of our experiments, we consider models in the low-data regime with large classification error rates, in which case, sample-wise agreement of two models is not of central importance. On the other hand, observe that if two models have population classification error rates less than $\epsilon/2$, then (2) is automatically satisfied, in which case, the notions of weak-approximation and strong-approximation converge.

We list several examples of $\epsilon$-approximation, with $\epsilon = 0$, for the case when $\tilde{A} = A$ are given by the following:

**Example 1: Support Vector Machines.** Given a dataset $D$ of size $N$, train an SVM on $D$ and obtain $M$ support vectors. These $M$ support vectors yield a dataset $\tilde{D}$ that is a strong 0-approximation to $D$ in the linearly separable case, while for the nonseparable case, one has to also include the datapoints with positive slack. Asymptotic lower bounds asserting $M = O(N)$ have been shown in Steinwart (2003).[2]

**Example 2: Ridge Regression.** Any two datasets $D$ and $\tilde{D}$ that determine the same ridge-regressor are 0-approximations of each other. In particular, in the scalar case, we can obtain arbitrarily small 0-approximating $\tilde{D}$ as follows. Given training data $D = (X, y)$ in $\mathbb{R}^d$, the corresponding ridge-regressor is the predictor

$$x^* \mapsto w \cdot x^*, \tag{3}$$

$$w = \Phi_\lambda(X) y, \tag{4}$$

$$\Phi_\lambda(X) = X^T (X X^T + \lambda I)^{-1} \tag{5}$$

---

[2] As a specific example, many thousands of support vectors are needed for MNIST classification (Bordes et al. (2005)).

where for $\lambda = 0$, we interpret the inverse as a pseudoinverse. It follows that for any given $w \in \mathbb{R}^{d \times 1}$, we can always find $(\tilde{X}, \tilde{y})$ of arbitrary size (i.e. $\tilde{X} \in \mathbb{R}^{n \times d}$, $y \in \mathbb{R}^{n \times 1}$ with $n$ arbitrarily small) that satisfies $w = \Phi_\lambda(\tilde{X})\tilde{y}$. Simply choose $\tilde{X}$ such that $w$ is in the range of $\Phi_\lambda(\tilde{X})$. The resulting dataset $(\tilde{X}, \tilde{y})$ is a 0-approximation to $D$. If we have a $C$-dimensional regression problem, the preceding analysis can be repeated component-wise in label-space to show 0-approximation with a dataset of size at least $C$ (since then the rank of $\Phi_\lambda(\tilde{X})$ can be made at least the rank of $w \in \mathbb{R}^{d \times C}$).

We are interested in learning algorithms given by KRR and neural networks. These can be investigated in unison via neural tangent kernels. Furthermore, we study two settings for the usage of $\epsilon$-approximate datasets, though there are bound to be others:

1. (Sample efficiency / compression) Fix $\epsilon$. What is the minimum size of $\tilde{D}$ needed in order for $\tilde{D}$ to be an $\epsilon$-approximate dataset?

2. (Privacy guarantee) Can an $\epsilon$-approximate dataset be found such that the distribution from which it is drawn and the distribution from which the original training dataset is drawn satisfy a given upper bound in mutual information?

Motivated by these questions, we introduce the following definitions:

**Definition 4.** (Heuristic) Let $\tilde{D}$ and $D$ be two datasets such that $\tilde{D}$ is a weak $\epsilon$-approximation of $D$, with $|\tilde{D}| \leq |D|$ and $\epsilon$ small. We call $|D|/|\tilde{D}|$ the *compression ratio*.

In other words, the compression ratio is a measure of how well $\tilde{D}$ compresses the information available in $D$, as measured by approximate agreement of their population loss. Our definition is heuristic in that $\epsilon$ is not precisely quantified and so is meant as a soft measure of compression.

**Definition 5.** Let $\Gamma$ be an algorithm that takes a dataset $D$ in $\mathbb{R}^d$ and returns a (random) collection of datasets in $\mathbb{R}^d$. For $0 \leq \rho \leq 1$, we say that $\Gamma$ is $\rho$-*corrupted* if for any input dataset $D$, every datapoint[3] drawn from the datasets of $\Gamma(D)$ has at least $\rho$ fraction of its coordinates independent of $D$.

In other words, datasets produced by $\Gamma$ have $\rho$ fraction of its entries contain no information about the dataset $D$ (e.g. because they have a fixed value or are filled in randomly). Corrupting information is naturally a way of enhancing privacy, as it makes it more difficult for an attacker to obtain useful information about the data used to train a model. Adding noise to the inputs to neural network or of its gradient updates can be shown to provide differentially private guarantees (Abadi et al. (2016)).

## 3 Kernel Inducing Points

Given a dataset $D$ sampled from a distribution $\mathcal{P}$, we want to find a small dataset $\tilde{D}$ that is an $\epsilon$-approximation to $D$ (or some large subset thereof) with respect to $(\tilde{A}, A, \ell, \mathcal{P})$. Focusing on $\tilde{A} = A$ for the moment, and making the approximation

$$\mathbb{E}_{(x,y)\in\mathcal{P}} \, \ell(\tilde{A}_{\tilde{D}}(x), y) \approx \mathbb{E}_{(x,y)\in D} \, \ell(\tilde{A}_{\tilde{D}}(x), y), \qquad (6)$$

this suggests we should optimize the right-hand side of (6) with respect to $\tilde{D}$, using $D$ as a validation set. For general algorithms $\tilde{A}$, the outer optimization for $\tilde{D}$ is computationally expensive and involves second-order derivatives, since one has to optimize over the inner loop encoded by the learning algorithm $\tilde{A}$. We are thus led to consider the class of algorithms drawn from kernel ridge-regression. The reason for this are two-fold. First, KRR performs convex-optimization resulting in a closed-form solution, so that when optimizing for the training parameters of KRR (in particular, the support data), we only have to consider first-order optimization. Second, since KRR for a neural tangent kernel (NTK) approximates the training of the corresponding wide neural network (Jacot et al., 2018; Lee et al., 2019; Arora et al., 2019a; Lee et al., 2020), we expect the use of neural kernels to yield $\epsilon$-approximations of $D$ for learning algorithms given by a broad class of neural networks trainings as well. (This will be validated in our experiments.)

---

[3]We ignore labels in our notion of $\rho$-corrupted since typically the label space has much smaller dimension than that of the datapoints.

---

**Algorithm 1:** Kernel Inducing Point (`KIP`)

---

**Require:** A target labeled dataset $(X_t, y_t)$ along with a kernel or family of kernels.
 1: Initialize a labeled support set $(X_s, y_s)$.
 2: **while** not converged **do**
 3:   Sample a random kernel. Sample a random batch $(\bar{X}_s, \bar{y}_s)$ from the support set. Sample a random batch $(\bar{X}_t, \bar{y}_t)$ from the target dataset.
 4:   Compute the kernel ridge-regression loss given by (7) using the sampled kernel and the sampled support and target data.
 5:   Backpropagate through $\bar{X}_s$ (and optionally $\bar{y}_s$ and any hyper-parameters of the kernel) and update the support set $(X_s, y_s)$ by updating the subset $(\bar{X}_s, \bar{y}_s)$.
 6: **end while**
 7: **return** Learned support set $(X_s, y_s)$

---

This leads to our first-order meta-learning algorithm `KIP` (Kernel Inducing Points), which uses kernel-ridge regression to learn $\epsilon$-approximate datasets. It can be regarded as an adaption of the inducing point method for Gaussian processes (Snelson & Ghahramani, 2006) to the case of KRR. Given a kernel $K$, the KRR loss function trained on a support dataset $(X_s, y_s)$ and evaluated on a target dataset $(X_t, y_t)$ is given by

$$L(X_s, y_s) = \frac{1}{2}\|y_t - K_{X_t X_s}(K_{X_s X_s} + \lambda I)^{-1} y_s\|^2, \tag{7}$$

where if $U$ and $V$ are sets, $K_{UV}$ is the matrix of kernel elements $(K(u, v))_{u \in U, v \in V}$. Here $\lambda > 0$ is a fixed regularization parameter. The `KIP` algorithm consists of optimizing (7) with respect to the support set (either just the $X_s$ or along with the labels $y_s$), see Algorithm 1. Depending on the downstream task, it can be helpful to use families of kernels (Step 3) because then `KIP` produces datasets that are $\epsilon$-approximations for a variety of kernels instead of a single one. This leads to a corresponding robustness for the learned datasets when used for neural network training. We remark on best experimental practices for sampling methods and initializations for `KIP` in the Appendix. Theoretical analysis for the convergence properties of `KIP` for the case of a linear kernel is provided by Theorem 1. Sample `KIP`-learned images can be found in Section F.

**KIP variations:** i) We can also randomly augment the sampled target batches in `KIP`. This effectively enhances the target dataset $(X_t, y_t)$, and we obtain improved results in this way, with no extra computational cost with respect to the support size. ii) We also can choose a corruption fraction $0 \le \rho < 1$ and do the following. Initialize a random $\rho$-percent of the coordinates of each support datapoint via some corruption scheme (zero out all such pixels or initialize with noise). Next, do not update such corrupted coordinates during the `KIP` training algorithm (i.e. we only perform gradient updates on the complementary set of coordinates). Call this resulting algorithm `KIP`$_\rho$. In this way, `KIP`$_\rho$ is $\rho$-corrupted according to Definition 5 and we use it to obtain our highly corrupted datasets.

**Label solving:** In addition to `KIP`, where we learn the support dataset via gradient descent, we propose another inducing point method, Label Solve (`LS`), in which we directly find the minimum of (7) with respect to the support labels while holding $X_s$ fixed. This is simple because the loss function is quadratic in $y_s$. We refer to the resulting labels

$$y_s^* = \Phi_0\Big(K_{X_t X_s}(K_{X_s X_s} + \lambda I)^{-1}\Big) y_t \tag{8}$$

as *solved labels*. As $\Phi_0$ is the pseudo-inverse operation, $y_s^*$ is the minimum-norm solution among minimizers of (7). If $K_{X_t X_s}$ is injective, using the fact that $\Phi_0(AB) = \Phi_0(B)\Phi_0(A)$ for $A$ injective and $B$ surjective (Greville (1966)), we can rewrite (8) as

$$y_s^* = (K_{X_s X_s} + \lambda I)\Phi_0(K_{X_t X_s}) y_t.$$

## 4  EXPERIMENTS

We perform three sets of experiments to validate the efficacy of `KIP` and `LS` for dataset learning. The first set of experiments investigates optimizing `KIP` and `LS` for compressing datasets and achieving

state of the art performance for individual kernels. The second set of experiments explores transferability of such learned datasets across different kernels. The third set of experiments investigate the transferability of `KIP`-learned datasets to training neural networks. The overall conclusion is that `KIP`-learned datasets, even highly corrupted versions, perform well in a wide variety of settings. Experimental details can be found in the Appendix.

We focus on MNIST (LeCun et al., 2010) and CIFAR-10 (Krizhevsky et al., 2009) datasets for comparison to previous methods. For `LS`, we also use Fashion-MNIST. These classification tasks are recast as regression problems by using mean-centered one-hot labels during training and by making class predictions via assigning the class index with maximal predicted value during testing. All our kernel-based experiments use the Neural Tangents library (Novak et al., 2020), built on top of JAX (Bradbury et al., 2018). In what follows, we use FC$m$ and Conv$m$ to denote a depth $m$ fully-connected or fully-convolutional network. Whether we mean a finite-width neural network or else the corresponding neural tangent kernel (NTK) will be understood from the context. We will sometimes also use the neural network Gaussian process (NNGP) kernel associated to a neural network in various places. By default, a neural kernel refers to NTK unless otherwise stated. RBF denotes the radial-basis function kernel. Myrtle-$N$ architecture follows that of Shankar et al. (2020), where an $N$-layer neural network consisting of a simple combination of $N-1$ convolutional layers along with $(2, 2)$ average pooling layers are inter-weaved to reduce internal patch-size.

We would have used deeper and more diverse architectures for `KIP`, but computational limits, which will be overcome in future work, placed restrictions, see the Experiment Details in Section D.

## 4.1 SINGLE KERNEL RESULTS

We apply `KIP` to learn support datasets of various sizes for MNIST and CIFAR-10. The objective is to distill the entire training dataset down to datasets of various fixed, smaller sizes to achieve high compression ratio. We present these results against various baselines in Tables 1 and 2. These comparisons occur cross-architecturally, but aside from Myrtle LS results, all our results involve the simplest of kernels (RBF or FC1), whereas prior art use deeper architectures (LeNet, AlexNet, ConvNet).

We obtain state of the art results for KRR on MNIST and CIFAR-10, for the RBF and FC1 kernels, both in terms of accuracy and number of images required, see Tables 1 and 2. In particular, our method produces datasets such that RBF and FC1 kernels fit to them rival the performance of deep convolutional neural networks on MNIST (exceeding 99.2%). By comparing Tables 2 and A2, we see that, e.g. 10 or 100 `KIP` images for RBF and FC1 perform on par with tens or hundreds times more natural images, resulting in a compression ratio of one or two orders of magnitude.

For neural network trainings, for CIFAR-10, the second group of rows in Table 2 shows that FC1 trained on `KIP` images outperform prior art, all of which have deeper, more expressive architectures. On MNIST, we still outperform some prior baselines with deeper architectures. This, along with the state of the art KRR results, suggests that `KIP`, when scaled up to deeper architectures, should continue to yield strong neural network performance.

For `LS`, we use a mix of NNGP kernels[4] and NTK kernels associated to FC1, Myrtle-5, Myrtle-10 to learn labels on various subsets of MNIST, Fashion-MNIST, and CIFAR-10. Our results comprise the bottom third of Tables 1 and 2 and Figure 2. As Figure 2 shows, the more targets are used, the better the performance. When all possible targets are used, we get an optimal compression ratio of roughly one order of magnitude at intermediate support sizes.

## 4.2 KERNEL TO KERNEL RESULTS

Here we investigate robustness of `KIP` and `LS` learned datasets when there is variation in the kernels used for training and testing. We draw kernels coming from FC and Conv layers of depths 1-3, since such components form the basic building blocks of neural networks. Figure A1 shows that `KIP`-datasets trained with random sampling of all six kernels do better on average than `KIP`-datasets trained using individual kernels.

---

[4]For FC1, NNGP and NTK perform comparably whereas for Myrtle, NNGP outperforms NTK.

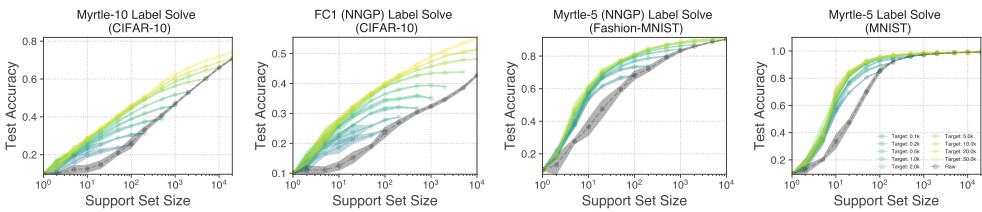

**Figure 2: LS performance for Myrtle-(5/10) and FC on CIFAR-10/Fashion-MNIST/MNIST.**
Results computed over 3 independent samples per support set size.

Table 1: **MNIST: KIP and LS vs baselines.** Comparing KRR (kernel ridge-regression) and NN (neural network) algorithms using various architectures and dataset distillation methods on datasets of varying sizes (10 to 10K).

| Alg. | Arch., Method | 10 | 100 | 500 | 5000 | 10000 |
|------|---------------|-----|-----|-----|------|-------|
| KRR | RBF, KIP | 89.60±0.09 | 97.31±0.09 | 98.29±0.06 | 98.70±0.04 | 98.74±0.04 |
| KRR | RBF, KIP (a + l)[1] | **90.63±0.27** | **97.84±0.06** | **98.85±0.04** | **99.31±0.04** | **99.34±0.03** |
| KRR | FC1, KIP | 89.30±0.01 | 96.64±0.08 | 97.64±0.06 | 98.52±0.04 | 98.59±0.05 |
| KRR | FC1, KIP (a + l) | 85.46±0.04 | 97.15±0.11 | 98.36±0.08 | 99.18±0.04 | 99.26±0.03 |
| NN | FC1, KIP[2] | 86.49±0.40 | 88.96±0.37 | 95.70±0.09 | 97.97±0.07 | - |
| NN | ConvNet,[3] DC[4] | **91.7±0.5** | **97.4±0.2** | - | - | - |
| NN | LeNet, DC | - | 93.9±0.6 | - | - | - |
| NN | LeNet, SLDD | - | 82.7±2.8 | - | - | - |
| NN | LeNet, DD | - | 79.5±8.1 | - | - | - |
| KRR | FC1, LS | 61.0±0.28 | 87.2±0.71 | 94.4±0.16 | 97.5±0.06 | 97.9±0.09 |
| KRR | Myrtle-5 NNGP, LS | **70.24±1.59** | **95.44±0.17** | **98.32±0.91** | **99.17±0.01** | **99.33±0.07** |
| KRR | Myrtle-5, LS | 68.50±2.52 | 95.53±0.22 | 98.17±0.07 | 99.05±0.06 | 99.22±0.02 |
| NN | LeNet, LD | 64.57±2.67 | 87.85±0.43 | 94.75±0.29 | - | - |

[1] (a + l) denotes KIP trained with augmentations and learning of labels

[2] KIP images are trained using the same kernel (FC1) corresponding to the evaluation neural network. Likewise for KRR, the train and test kernels coincide.

[3] ConvNet is neural network consisting of 3 convolutional blocks, where a block consists of convolution, instance normalization, and a (2,2) average pooling. See Zhao et al. (2020).

[4] DC (Zhao et al., 2020), LD (Bohdal et al., 2020), SLDD (Sucholutsky & Schonlau, 2019), DD (Wang et al., 2018).

For LS , transferability between FC1 and Myrtle-10 kernels on CIFAR-10 is highly robust, see Figure A2. Namely, one can label solve using FC1 and train Myrtle-10 using those labels and vice versa. There is only a negligible difference in performance in nearly all instances between data with transferred learned labels and with natural labels.

## 4.3 KERNEL TO NEURAL NETWORKS RESULTS

Significantly, KIP -learned datasets, even with heavy corruption, transfer remarkably well to the training of neural networks. Here, corruption refers to setting a random $\rho$ fraction of the pixels of each image to uniform noise between $-1$ and $1$ (for KIP , this is implemented via $KIP_\rho$)[5]. The deterioriation in test accuracy for KIP -images is limited as a function of the corruption fraction, especially when compared to natural images, and moreover, corrupted KIP -images typically outperform *uncorrupted* natural images. We verify these conclusions along the following dimensions:

**Robustness to dataset size:** We perform two sets of experiments.

(i) First, we consider small KIP datasets (10, 100, 200 images) optimized using multiple kernels (FC1-3, Conv1-2), see Tables A5, A6. We find that our in-distribution transfer (the downstream neural network has its neural kernel included among the kernels sampled by KIP ) performs re-

[5]Our images are preprocessed so as to be mean-centered and unit-variance per pixel. This choice of corruption, which occurs post-processing, is therefore meant to (approximately) match the natural pixel distribution.

Table 2: **CIFAR-10: KIP and LS vs baselines.** Comparing KRR (kernel ridge-regression) and NN (neural network) algorithms using various architectures and dataset distillation methods on datasets of various sizes (10 to 10K). Notation same as in Table 1.

| Alg. | Arch., Method | 10 | 100 | 500 | 5000 | 10000 |
|------|---------------|------|------|------|------|-------|
| KRR | RBF, KIP | 39.9±0.9 | 49.3±0.3 | 51.2±0.8 | - | - |
| KRR | RBF, KIP (a + l) | 40.3±0.5 | **53.8±0.3** | **60.1±0.2** | **65.6±0.2** | **66.3±0.2** |
| KRR | FC1, KIP | 39.3±1.6 | 49.1±1.1 | 52.1±0.8 | 54.5±0.5 | 54.9±0.5 |
| KRR | FC1, KIP (a + l) | **40.5±0.4** | 53.1±0.5 | 58.6±0.4 | 63.8±0.3 | 64.6±0.2 |
| NN | FC1, KIP | **36.2±0.1** | **45.7±0.3** | 46.9±0.2 | 50.1±0.4 | 51.7±0.4 |
| NN | ConvNet, DC | 28.3±0.5 | 44.9±0.5 | - | - | - |
| NN | AlexNet, DC | - | 39.1±1.2 | - | - | - |
| NN | AlexNet, SLDD | - | 39.8±0.8 | - | - | - |
| NN | AlexNet, DD | - | 36.8±1.2 | - | - | - |
| KRR | FC1 NNGP, LS | 27.5±0.3 | 40.1±0.3 | 46.4±0.4 | 53.5±0.2 | 55.1±0.3 |
| KRR | Myrtle-10 NNGP, LS + ZCA[5] | **31.7±0.2** | **56.0±0.5** | **69.8±0.1** | **80.2±0.1** | **82.3±0.1** |
| KRR | Myrtle-10, LS | 28.8±0.4 | 45.8±0.7 | 58.0±0.3 | 69.6±0.2 | 72.0±0.2 |
| NN | AlexNet, LD | 25.69±0.72 | 38.33±0.44 | 43.16±0.47 | - | - |

[5] We apply regularized ZCA whitening instead of standard preprocessing to the images, see Appendix D for further details.

markably well, with both uncorrupted *and* corrupted `KIP` images beating the uncorrupted natural images of corresponding size. Out of distribution networks (LeNet (LeCun et al., 1998) and Wide Resnet (Zagoruyko & Komodakis, 2016)) have less transferability: the uncorrupted images still outperform natural images, and corrupted `KIP` images still outperform corrupted natural images, but corrupted `KIP` images no longer outperform uncorrupted natural images.

(ii) We consider larger `KIP` datasets (1K, 5K, 10K images) optimized using a single FC1 kernel for training of a corresponding FC1 neural network, where the `KIP` training uses augmentations (with and without label learning), see Tables A7-A10 and Figure A3. We find, as before, `KIP` images outperform natural images by an impressive margin: for instance, on CIFAR-10, 10K `KIP` -learned images with 90% corruption achieves 49.9% test accuracy, exceeding 10K natural images with no corruption (acc: 45.5%) and 90% corruption (acc: 33.8%). Interestingly enough, sometimes higher corruption leads to *better* test performance (this occurs for CIFAR-10 with cross entropy loss for both natural and `KIP` -learned images), a phenomenon to be explored in future work. We also find that `KIP` with label-learning often tends to harm performance, perhaps because the labels are overfitting to KRR.

**Robustness to hyperparameters:** For CIFAR-10, we took 100 images, both clean and 90% corrupted, and trained networks on a wide variety of hyperparameters for various neural architectures. We considered both neural networks whose corresponding neural kernels were sampled during `KIP` -training those that were not. We found that in both cases, the `KIP` -learned images almost always outperform 100 random natural images, with the optimal set of hyperparameters yielding a margin close to that predicted from the KRR setting, see Figure 3. This suggests that `KIP` -learned images can be useful in accelerating hyperparameter search.

## 5 RELATED WORK

**Coresets:** A classical approach for compressing datasets is via subset selection, or some approximation thereof. One notable work is Borsos et al. (2020), utilizing KRR for dataset subselection. For an overview of notions of coresets based on pointwise approximatation of datasets, see Phillips (2016).

**Neural network approaches to dataset distillation:** Maclaurin et al. (2015); Lorraine et al. (2020) approach dataset distillation through learning the input images from large-scale gradient-based meta-learning of hyper-parameters. Properties of distilled input data was first analyzed in Wang et al. (2018). The works Sucholutsky & Schonlau (2019); Bohdal et al. (2020) build upon Wang et al.

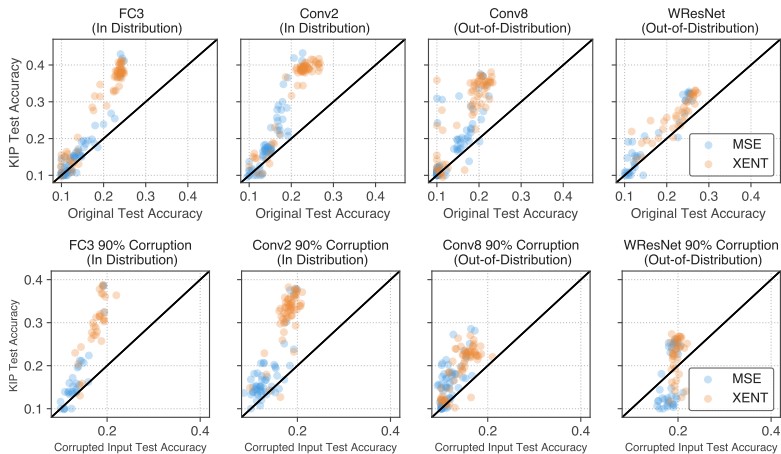

Figure 3: **KIP learned images transfers well to finite neural networks.** Test accuracy on CIFAR-10 comparing natural images (x-axis) and `KIP`-learned images (y-axis). Each scatter point corresponds to varying hyperparameters for training (e.g. learning rate). Top row are clean images, bottom row are 90% corrupted images. `KIP` images were trained using FC1-3, Conv1-2 kernels.

(2018) by distilling labels. More recently, Zhao et al. (2020) proposes condensing training set by gradient matching condition and shows improvement over Wang et al. (2018).

**Inducing points:** Our approach has as antecedant the inducing point method for Gaussian Processes (Snelson & Ghahramani, 2006; Titsias, 2009). However, whereas the latter requires a probabilistic framework that optimizes for marginal likelihood, in our method we only need to consider minimizing mean-square loss on validation data.

**Low-rank kernel approximations:** Unlike common low-rank approximation methods (Williams & Seeger, 2001; Drineas & Mahoney, 2005), we obtain not only a low-rank support-support kernel matrix with `KIP`, but also a low-rank target-support kernel matrix. Note that the resulting matrices obtained from `KIP` need not approximate the original support-support or target-support matrices since `KIP` only optimizes for the loss function.

**Neural network kernels:** Our work is motivated by the exact correspondence between infinitely-wide neural networks and kernel methods (Neal, 1994; Lee et al., 2018; Matthews et al., 2018; Jacot et al., 2018; Novak et al., 2019; Garriga-Alonso et al., 2019; Arora et al., 2019a). These correspondences allow us to view both Bayesian inference and gradient descent training of wide neural networks with squared loss as yielding a Gaussian process or kernel ridge regression with neural kernels.

**Instance-Based Encryption:** A related approach to corrupting datasets involves encrypting individual images via sign corruption (Huang et al. (2020)).

## 6 CONCLUSION

We introduced novel algorithms `KIP` and `LS` for the meta-learning of datasets. We obtained a variety of compressed and corrupted datasets, achieving state of the art results for KRR and neural network dataset distillation methods. This was achieved even using the simplest of kernels and neural networks (shallow fully-connected networks and purely-convolutional networks without pooling), which notwithstanding their limited expressiveness, outperform most baselines that use deeper architectures. Follow-up work will involve scaling up `KIP` to deeper architectures with pooling (achievable with multi-device training) for which we expect to obtain even more highly performant datasets, both in terms of overall accuracy and architectural flexibility. Finally, we obtained highly corrupt datasets whose performance match or exceed natural images, which when developed at scale, could lead to practical applications for privacy-preserving machine learning.

ACKNOWLEDGMENTS

We would like to thank Dumitru Erhan, Yang Li, Hossein Mobahi, Jeffrey Pennington, Si Si, Jascha Sohl-Dickstein, and Lechao Xiao for helpful discussions and references.

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

## A    REMARKS ON DEFINITION OF $\epsilon$-APPROXIMATION

Here, we provide insights into the formulation of Definition 3. One noticeable feature of our definition is that it allows for different algorithms $A$ and $\tilde{A}$ when comparing datasets $D$ and $\tilde{D}$. On the one hand, such flexibility is required, since for instance, a mere preprocessing of the dataset (e.g. rescaling it), should be regarded as producing an equivalent (0-approximate) dataset. Yet such a rescaling may affect the hyperparameters needed to train an equivalent model (e.g. the learning rate). Thus, one must allow the relevant hyperparameters of an algorithm to vary when the datasets are also varying. On the other hand, it would be impossible to compare two datasets meaningfully if the learned algorithms used to train them differ too significantly. For instance, if $D$ is a much larger dataset than $\tilde{D}$, but $A$ is a much less expressive algorithm than $\tilde{A}$, then the two datasets may be $\epsilon$-approximations of each other, but it would be strange to compare $D$ and $\tilde{D}$ in this way. Thus, we treat the notion of what class of algorithms to consider informally, and leave its specification as a practical matter for each use case. In practice, the pair of algorithms we use to compare datasets should be drawn from a family in which some reasonable range of hyperparameters are varied, the ones typically tuned when learning on an unknown dataset. The main case for us with differing $A$ and $\tilde{A}$ is when we compare neural network training alongside kernel ridge-regression.

Another key feature of our definition is that datapoints of an $\epsilon$-approximating dataset must have the same shape as those of the original dataset. This makes our notion of an $\epsilon$-approximate dataset more restrictive than returning a specialized set of extracted features from some initial dataset.

Analogues of our $\epsilon$-approximation definition have been formulated in the unsupervised setting, e.g. in the setting of clustering data (Phillips, 2016; Jubran et al., 2019).

Finally, note that the loss function $\ell$ used for comparing datasets does not have to coincide with any loss functions optimized in the learning algorithms $A$ and $\tilde{A}$. Indeed, for kernel ridge-regression, training mimimizes mean square loss while $\ell$ can be 0-1 loss.

## B    TUNING KIP

**Sampling:**  When optimizing for KRR performance with support dataset size $N$, it is best to learn a support set $\tilde{D}$ of size $N$ and sample this entire set during KIP training. It is our observation that subsets of size $M < N$ of $\tilde{D}$ will not perform as well as optimizing directly for a size $M$ dataset through KIP . Conversely, sampling subsets of size $M$ from a support dataset of size $N$ during KIP will not lead to a dataset that does as well as optimizing for all $N$ points. This is sensible: optimizing for small support size requires a resourceful learning of coarse features at the cost of learning fine-grained features from many support datapoints. Conversely, optimizing a large support set means the learned support set has leveraged higher-order information, which will degrade when restricted to smaller subsets.

For sampling from the target set, which we always do in a class-balanced way, we found larger batch sizes typically perform better on the test set if the train and test kernels agree. If the train and test kernels differ, then smaller batch sizes lead to less overfitting to the train kernel.

**Initialization:**  We tried two sets of initializations. The first ("image init") initializes $(X_s, y_s)$ to be a subset of $(X_t, y_t)$. The second ("noise init") initializes $X_s$ with uniform noise and $y_s$ with mean-centered, one-hot labels (in a class-balanced way). We found image initialization to perform better.

**Regularization:** The regularization parameter $\lambda$ in (7) can be replaced with $\frac{1}{n}\lambda \cdot \operatorname{tr}(K_{X_s X_s})$, where $n$ is the number of datapoints in $X_s$. This makes the loss function invariant with respect to rescaling of the kernel function $K$ and also normalizes the regularization with respect to support size. In practice, we use this scale-invariant regularization with $\lambda = 10^{-6}$.

**Number of Training Iterations:** Remarkably, KIP converges very quickly in all experimental settings we tried. After only on the order of a hundred iterations, independently of the support size, kernel, and corruption factor, the learned support set has already undergone the majority of its learning (test accuracy is within more than 90% of the final test accuracy). For the platforms available to us, using a single V100 GPU, one hundred training steps for the experiments we ran involving

target batch sizes that were a few thousand takes on the order of about 10 minutes. When we add augmentations to our targets, performance continues to improve slowly over time before flattening out after several thousands of iterations.

## C  THEORETICAL RESULTS

Here, we analyze convergence properties of KIP in returning an $\epsilon$-approximate dataset. In what follows, we refer to gradient-descent KIP as the case when we sample from the entire support and train datasets for each update step to KIP . We also assume that the distribution $\mathcal{P}$ used to evaluate $\epsilon$-approximation is supported on inputs $x \in \mathbb{R}^d$ with $\|x\| \leq 1$ (merely to provide a convenient normalization when evaluating loss on regression algorithms).

For the case of a linear kernel, we prove the below convergence theorem:

**Theorem 1.** *Let $D = (X_t, y_t) \in \mathbb{R}^{n_t \times d} \times \mathbb{R}^{n_t \times C}$ be an arbitrary dataset. Let $w_\lambda \in \mathbb{R}^{d \times C}$ be the coefficients obtained from training $\lambda$ ridge-regression ($\lambda$-RR) on $(X_t, y_t)$, as given by (4).*

1. *For generic[6] initial conditions for the support set $(X_s, y_s) \subset \mathbb{R}^{n_s \times d} \times \mathbb{R}^{n_s \times C}$ and sufficiently small $\lambda > 0$, gradient descent KIP with target dataset $D$ converges to a dataset $\tilde{D}$.*

2. *The dataset $\tilde{D}$ is a strong $\epsilon$-approximation to $D$ with respect to algorithms ($\lambda$-RR, 0-RR) and loss function equal to mean-square loss, where*

$$\epsilon \leq \frac{1}{2}\|\tilde{w} - w_0\|_2^2 \tag{A1}$$

*and $\tilde{w} \in \mathbb{R}^{d \times C}$ are the coefficients of the linear classifier obtained from training $\lambda$-RR on $\tilde{D}$. If the size of $\tilde{D}$ is at least $C$, then $\tilde{w}$ is also a least squares classifier for $D$. In particular, if $D$ has a unique least squares classifier, then $\epsilon = 0$.*

*Proof.* We discuss the case where $X_s$ is optimized, with the case where both $(X_s, y_s)$ are optimized proceeding similarly. In this case, by genericity, we can assume $y_s \neq 0$, else the learning dynamics is trivial. Furthermore, to simplify notation for the time being, assume the dimensionality of the label space is $C = 1$ without loss of generality. First, we establish convergence. For a linear kernel, we can write our loss function as

$$L(X_s) = \frac{1}{2}\|y_t - X_t X_s^T (X_s X_s^T + \lambda I)^{-1} y_s\|^2, \tag{A2}$$

defined on the space $\mathbb{M}_{n_s \times d}$ of $n_s \times d$ matrices. It is the pullback of the loss function

$$L_{\mathbb{R}^{d \times n_s}}(\Phi) = \frac{1}{2}\|y_t - X_t \Phi y_s\|^2, \qquad \Phi \in \mathbb{R}^{d \times n_s} \tag{A3}$$

under the map $X_s \mapsto \Phi_\lambda(X_s) = X_s^T (X_s X_s^T + \lambda I)^{-1}$. The function (A3) is quadratic in $\Phi$ and all its local minima are global minima given by an affine subspace $\mathcal{M} \subset \mathbb{M}_{d \times n_s}$. Moreover, each point of $\mathcal{M}$ has a stable manifold of maximal dimension equal to the codimension of $\mathcal{M}$. Thus, the functional $L$ has global minima given by the inverse image $\Phi_\lambda^{-1}(\mathcal{M})$ (which will be nonempty for sufficiently small $\lambda$).

Next, we claim that given a fixed initial $(X_s, y_s)$, then for sufficiently small $\lambda$, gradient-flow of (A2) starting from $(X_s, y_s)$ cannot converge to a non-global local minima. We proceed as follows. If $X = U\Sigma V^T$ is a singular value decomposition of $X$, with $\Sigma$ a $n_s \times n_s$ diagonal matrix of singular values (and any additional zeros for padding), then $\Phi(X) = V\phi(\Sigma)U^T$ where $\phi(\Sigma)$ denotes the diagonal matrix with the map

$$\phi : \mathbb{R}^{\geq 0} \to \mathbb{R}^{\geq 0} \tag{A4}$$

$$\phi(\mu) = \frac{\mu}{\mu^2 + \lambda} \tag{A5}$$

---

[6]A set can be generic by either being open and dense or else having probability one with respect to some measure absolutely continuous with respect to Lebesgue measure. In our particular case, generic refers to the complement of a submanifold of codimension at least one.

applied to each singular value of $\Sigma$. The singular value decomposition depends analytically on $X$ (Kato (1976)). Given that $\phi : \mathbb{R}^{\geq 0} \to \mathbb{R}^{\geq 0}$ is a local diffeomorphism away from its maximum value at $\mu = \mu^* := \lambda^{1/2}$, it follows that $\Phi_\lambda : \mathbb{M}_{n_s \times d} \to \mathbb{M}_{d \times n_s}$ is locally surjective, i.e. for every $X$, there exists a neighborhood $\mathcal{U}$ of $X$ such that $\Phi_\lambda(\mathcal{U})$ contains a neighborhood of $\Phi_\lambda(X)$. Thus, away from the locus of matrices in $\mathbb{M}_{n_s \times d}$ that have a singular value equaling $\mu^*$, the function (A2) cannot have any non-global local minima, since the same would have to be true for (A3). We are left to consider those matrices with some singular values equaling $\mu^*$. Note that as $\lambda \to 0$, we have $\phi(\mu^*) \to \infty$. On the other hand, for any initial choice of $X_s$, the matrices $\Phi_\lambda(X_s)$ have uniformly bounded singular values as a function of $\lambda$. Moreover, as $X_s = X_s(t)$ evolves, $\|\Phi_\lambda(X_s(t))\|$ never needs to be larger than some large constant times $\|\Phi_\lambda(X_s(0))\| + \frac{\|y_t\|}{\mu^+ \|y_s\|}$, where $\mu_+$ is the smallest positive singular value of $X_t$. Consequently, $X_s(t)$ never visits a matrix with singular value $\mu^*$ for sufficiently small $\lambda > 0$; in particular, we never have to worry about convergence to a non-global local minimum.

Thus, a generic gradient trajectory $\gamma$ of $L$ will be such that $\Phi_\lambda \circ \gamma$ is a gradient-like[7] trajectory for $L_{\mathbb{R}^{d \times n_s}}$ that converges to $\mathcal{M}$. We have to show that $\gamma$ itself converges. It is convenient to extend $\phi$ to a map defined on the one-point compactification $[0, \infty] \supset \mathbb{R}^{\geq 0}$, so as to make $\phi$ a two-to-one map away from $\mu^*$. Applying this compactification to every singular value, we obtain a compactification $\overline{\mathbb{M}}^{n_s \times d}$ of $\mathbb{M}^{n_s \times d}$, and we can naturally extend $\Phi_\lambda$ to such a compactification. We have that $\gamma$ converges to an element of $\tilde{\mathcal{M}} := \Phi_\lambda^{-1}(\mathcal{M}) \subset \overline{\mathbb{M}}^{n_s \times d}$, where we need the compactification to account for the fact that when $\Phi_\lambda \circ \gamma$ converges to a matrix that has a zero singular value, $\gamma$ may have one of its singular values growing to infinity. Let $\mathcal{M}_0$ denote the subset of $\mathcal{M}$ with a zero singular value. Then $\gamma$ converges to an element of $\mathbb{M}^{n_s \times d}$ precisely when $\gamma$ does not converge to an element of $\tilde{\mathcal{M}}_\infty := \Phi_\lambda^{-1}(\mathcal{M}_0) \cap (\overline{\mathbb{M}}^{n_s \times d} \setminus \mathbb{M}^{n_s \times d})$. However, $\mathcal{M}_0 \subset \mathcal{M}$ has codimension one and hence so does $\tilde{\mathcal{M}}_\infty \subset \Phi_\lambda^{-1}(\mathcal{M}_0)$. Thus, the stable set to $\tilde{\mathcal{M}}_\infty$ has codimension one in $\overline{\mathbb{M}}^{n_s \times d}$, and hence its complement is nongeneric. Hence, we have generic convergence of a gradient trajectory of $L$ to a (finite) solution. This establishes the convergence result of Part 1.

For Part 2, the first statement is a general one: the difference of any two linear models, when evaluated on $\mathcal{P}$, can be pointwise bounded by the spectral norm of the difference of the model coefficient matrices. Thus $\tilde{D}$ is a strong $\epsilon$-approximation to $D$ with respect to ($\lambda$-RR, 0-RR) where $\epsilon$ is given by (A1). For the second statement, observe that $L$ is also the pullback of the loss function

$$L_{\mathbb{R}^{d \times C}}(w) = \frac{1}{2}\|y_t - X_t w\|^2, \qquad w \in \mathbb{R}^{d \times C}. \tag{A6}$$

under the map $X_s \mapsto w(X_s) = \Phi_\lambda(X_s)y_s$. The function $L_{\mathbb{R}^{d \times C}}(w)$ is quadratic in $w$ and has a unique minimum value, with the space of global minima being an affine subspace $W^*$ of $\mathbb{R}^d$ given by the least squares classifiers for the dataset $(X_t, y_t)$. Thus, the global minima of $L$ are the preimage of $W^*$ under the map $w(X_s)$. For generic initial $(X_s, y_s)$, we have $y_s \in \mathbb{R}^{n_s \times C}$ is full rank. This implies, for $n_s \geq C$, that the range of all possible $w(X_s)$ for varying $X_s$ is all of $\mathbb{R}^{d \times C}$, so that the minima of (A6) and (A3) coincide. This implies the final parts of Part 2. $\qquad \square$

We also have the following result about $\epsilon$-approximation using the label solve algorithm:

**Theorem 2.** *Let $D = (X_t, y_t) \in \mathbb{R}^{n_t \times d} \times \mathbb{R}^{n_t \times C}$ be an arbitrary dataset. Let $w_\lambda \in \mathbb{R}^{d \times C}$ be the coefficients obtained from training $\lambda$ ridge-regression ($\lambda$-RR) on $(X_t, y_t)$, as given by (4). Let $X_s \in \mathbb{R}^{n_s \times d}$ be an arbitrary initial support set and let $\lambda \geq 0$. Define $y_s^* = y_s^*(\lambda)$ via (8).*

*Then $(X_s, y_s^*)$ yields a strong $\epsilon(\lambda)$-approximation of $(X_t, y_t)$ with respect to algorithms ($\lambda$-RR, 0-RR) and mean-square loss, where*

$$\epsilon(\lambda) = \frac{1}{2}\|w^*(\lambda) - w_0\|_2^2 \tag{A7}$$

*and $w^*(\lambda)$ is the solution to*

$$w^*(\lambda) = \mathrm{argmin}_{w \in W} \|y_t - X_t w\|^2, \qquad W = \mathrm{im}\left(\Phi_\lambda(X_s) : \ker\left(X_t \Phi_\lambda(X_s)\right)^\perp \to \mathbb{R}^{d \times C}\right). \tag{A8}$$

---

[7] A vector field $v$ is gradient-like for a function $f$ if $v \cdot \mathrm{grad}(f) \geq 0$ everywhere.

*Moreover, for $\lambda = 0$, if $\mathrm{rank}(X_s) = \mathrm{rank}(X_t) = d$, then $w^*(\lambda) = w_0$. This implies $y_s^* = X_s w_0$, i.e. $y_s^*$ coincides with the predictions of the 0-RR classifier trained on $(X_t, y_t)$ evaluated on $X_s$.*

*Proof.* By definition, $y_s^*$ is the minimizer of

$$L(y_s) = \frac{1}{2}\|y_t - X_t \Phi_\lambda(X_s) y_s\|^2,$$

with minimum norm. This implies $y_s^* \in \ker\left(X_t \Phi_\lambda(X_s)\right)^\perp$ and that $w^*(\lambda) = \Phi_\lambda(X_s) y_s^*$ satisfies (A8). At the same time, $w^*(\lambda) = \Phi_\lambda(X_s) y_s^*$ are the coefficients of the $\lambda$-RR classifier trained on $(X_s, y_s^*)$. If $\mathrm{rank}(X_s) = \mathrm{rank}(X_t) = d$, then $\Phi_0(X_s)$ is surjective and $X_t$ is injective, in which case

$$\begin{aligned}
\omega^*(0) &= \Phi_0(X_s) y_s^* \\
&= \Phi_0(X_s)\Phi_0(X_t \Phi_0(X_s)) y_t \\
&= \Phi_0(X_s) X_s \Phi_0(X_t) y_t \\
&= \omega_0.
\end{aligned}$$

The results follow. $\square$

For general kernels, we make the following simple observation concerning the optimal output of `KIP`.

**Theorem 3.** *Fix a target dataset $(X_t, y_t)$. Consider the family of all subspaces $\mathcal{S}$ of $\mathbb{R}^{n_t}$ given by $\{\mathrm{im}\, K_{X_t X_s} : X_s \in \mathbb{R}^{n_s \times d}\}$, i.e. all possible column spaces of $K_{X_t X_s}$. Then the infimum of the loss (7) over all possible $(X_s, y_s)$ is equal to $\inf_{S \in \mathcal{S}} \frac{1}{2}\|\Pi_S^\perp y_t\|^2$ where $\Pi_S^\perp$ is orthogonal projection onto the orthogonal complement of $S$ (acting identically on each label component).*

*Proof.* Since $y_s$ is trainable, $(K_{X_s X_s} + \lambda)^{-1} y_s$ is an arbitrary vector in $\mathbb{R}^{n_s \times C}$. Thus, minimizing the training objective corresponds to maximizing the range of the linear map $K_{X_t X_s}$ over all possible $X_s$. The result follows. $\square$

## D  EXPERIMENT DETAILS

In all `KIP` trainings, we used the Adam optimizer. All our labels are mean-centered 1-hot labels. We used learning rates $0.01$ and $0.04$ for the MNIST and CIFAR-10 datasets, respectively. When sampling target batches, we always do so in a class-balanced way. When augmenting data, we used the ImageGenerator class from Keras, which enables us to add horizontal flips, height/width shift, rotatations (up to 10 degrees), and channel shift (for CIFAR-10). All datasets are preprocessed using channel-wise standardization (i.e. mean subtraction and division by standard-deviation). For neural (tangent) kernels, we always use weight and bias variance $\sigma_w^2 = 2$ and $\sigma_b^2 = 10^{-4}$, respectively. For both neural kernels and neural networks, we always use ReLU activation. Convolutional layers all use a $(3, 3)$ filter with stride 1 and same padding.

*Compute Limitations:* Our neural kernel computations, implemented using Neural Tangents libraray (Novak et al., 2020) are such that computation scales (i) linearly with depth; (ii) quadratically in the number of pixels for convolutional kernels; (iii) quartically in the number of pixels for pooling layers. Such costs mean that, using a single V100 GPU with 16GB of RAM, we were (i) only able to sample shallow kernels; (ii) for convolutional kernels, limited to small support sets and small target batch sizes; (iii) unable to use pooling if learning more than just a few images. Scaling up `KIP` to deeper, more expensive architectures, achievable using multi-device training, will be the subject of future exploration.

*Kernel Parameterization:* Neural tangent kernels, or more precisely each neural network layer of such kernels, as implemented in Novak et al. (2020) can be parameterized in either the "NTK" parameterization or "standard" parameterization Sohl-Dickstein et al. (2020). The latter depends on the width of a corresponding finite-width neural network while the former does not. Our experiments mix both these parameterizations for variety. However, because we use a scale-invariant

regularization for KRR (Section B), the choice of parameterization has a limited effect compared to other more significant hyperparameters (e.g. the support dataset size, learning rate, etc.)[8].

*Single kernel results:* (Tables 1 and 2) For FC, we used kernels with NTK parametrization. For RBF, our rbf kernel is given by

$$\mathrm{rbf}(x_1, x_2) = \exp(-\gamma \|x_1 - x_2\|^2 / d) \tag{A9}$$

where $d$ is the dimension of the inputs and $\gamma = 1$. We found that treating $\gamma$ as a learnable parameter during `KIP` had mixed results[9] and so keep it fixed for simplicity.

For MNIST, we found target batch size equal to 6K sufficient. For CIFAR-10, it helped to sample the entire training dataset of 50K images per step (hence, along with sampling the full support set, we are doing full gradient descent training). When support dataset size is small or if augmentations are employed, there is no overfitting (i.e. the train and test loss/accuracy stay positively correlated). If the support dataset size is large (5K or larger), sometimes there is overfitting when the target batch size is too large (e.g. for the RBF kernel on CIFAR10, which is why we exclude in Table 2 the entries for 5K and 10K). We could have used a validation dataset for a stopping criterion, but that would have required reducing the target dataset from the entire training dataset.

We train `KIP` for 10-20k iterations and took 5 random subsets of images for initializations. For each such training, we took 5 checkpoints with lowest train and loss and computed the test accuracy. This gives 25 evaluations, for which we can compute the mean and standard deviation for our test accuracy numbers in Tables 1 and 2.

*Kernel transfer results:* For transfering of `KIP` images, both to other kernels and to neural networks, we found it useful to use smaller target batch sizes (either several hundred or several thousand), else the images overfit to their source kernel. For random sampling of kernels used in Figure A1 and producing datasets for training of neural networks, we used FC kernels with width 1024 and Conv kernels with width 128, all with standard parametrization.

*Neural network results:* Neural network trainings on natural data with mean-square loss use mean-centered one-hot labels for consistency with `KIP` trainings. For cross entropy loss, we use one-hot labels. For neural network trainings on `KIP` -learned images with label learning, we transfer over the labels directly (as with the images), whatever they may be.

For neural network transfer experiments occurring in Table 1, Table 2, Figure 3, Table A5, and Table A6, we did the following. First, the images were learned using kernels FC1-3, Conv1-2. Second, we trained for a few hundred iterations, after which optimal test performance was achieved. On MNIST images, we trained the networks with constant learning rate and Adam optimizer with cross entropy loss. Learning rate was tuned over small grid search space. For the FC kernels and networks, we use width of 1024. On CIFAR-10 images, we trained the networks with constant learning rate, momentum optimizer with momentum 0.9. Learning rate, L2 regularization, parameterization (standard vs NTK) and loss type (mean square, softmax-cross-entropy) was tuned over small grid search space. Vanilla networks use constant width at each layer: for FC we use width of 1024, for Conv2 we use 512 channels, and for Conv8 we use 128 channels. No pooling layers are used except for the WideResNet architecture, where we follow the original architecture of Zagoruyko & Komodakis (2016) except that our batch normalization layer is stateless (i.e. no exponential moving average of batch statistics are recorded).

For neural network transfer experiments in Figure A3, Tables A7-A10, we did the following. Our `KIP` -learned images were trained using only an FC1 kernel. The neural network FC1 has an increased width 4096, which helps with the larger number of images. We used learning rate $4 \times 10^{-4}$ and the Adam optimizer. The `KIP` learned images with only augmentations used target batch size equal to half the training dataset size and were trained for 10k iterations, since the use of augmentations allows for continued gains after longer training. The `KIP` learned images with augmentations

---

[8]All our final readout layers use the fixed NTK parameterization and all our statements about which parameterization we are using should be interpreted accordingly. This has no effect on the training of our neural networks while for kernel results, this affects the recursive formula for the NTK at the final layer if using standard parameterization (by the changing the relative scales of the terms involved). Since the train/test kernels are consistently parameterized and `KIP` can adapt to the scale of the kernel, the difference between our hybrid parameterization and fully standard parameterization has a limited affect.

[9]On MNIST it led to very slight improvement. For CIFAR10, for small support sets, the effect was a small improvement on the test set, whereas for large support sets, we got worse performance.

Table A1: **Accuracy on random subsets of MNIST.** Standard deviations over 20 resamplings.

| # Images \ Kernel | Linear | RBF | FC1 |
|---|---|---|---|
| 10 | 44.6±3.7 | 45.3±3.9 | 45.8±3.9 |
| 20 | 51.9±3.1 | 54.6±2.9 | 54.7±2.8 |
| 40 | 59.4±2.4 | 66.9±2.0 | 66.0±1.9 |
| 80 | 62.6±2.7 | 75.6±1.6 | 74.3±1.7 |
| 160 | 62.2±2.1 | 82.7±1.4 | 81.1±1.6 |
| 320 | 52.3±1.9 | 88.1±0.8 | 86.9±0.9 |
| 640 | 41.9±1.4 | 91.8±0.5 | 91.1±0.5 |
| 1280 | 71.0±0.9 | 94.2±0.3 | 93.6±0.3 |
| 2560 | 79.7±0.5 | 95.7±0.2 | 95.3±0.2 |
| 5000 | 83.2±0.4 | 96.8±0.2 | 96.4±0.2 |
| 10000 | 84.9±0.4 | 97.5±0.2 | 97.2±0.2 |

and label learning used target batch size equal to a tenth of the training dataset size and were trained for 2k iterations (the learned data were observed to overfit to the kernel and have less transferability if larger batch size were used or if trainings were carried out longer).

All neural network trainings were run with 5 random initializations to compute mean and standard deviation of test accuracies.

In Table 2, regularized ZCA preprocessing was used for a Myrtle-10 kernel (denoted with *ZCA*) on CIFAR-10 dataset. Shankar et al. (2020) and Lee et al. (2020) noticed that for neural (convolutional) kernels on image classification tasks, regularized ZCA preprocessing can improve performance significantly compared to standard preprocessing. We follow the prepossessing scheme used in Shankar et al. (2020), with regularization strength of $10^{-5}$ without augmentation.

# E   TABLES AND FIGURES

## E.1   KERNEL BASELINES

We report various baselines of KRR trained on natural images. Tables A1 and A2 shows how various kernels vary in performance with respect to random subsets of MNIST and CIFAR-10. Linear denotes a linear kernel, RBF denotes the rbf kernel (A9) with $\gamma = 1$, and FC1 uses standard parametrization and width 1024. Interestingly enough, we observe non-monotonicity for the linear kernel, owing to double descent phenomenon Hastie et al. (2019). We include additional columns for deeper kernel architectures in Table A2, taken from Shankar et al. (2020) for reference.

Comparing Tables 1, 2 with Tables A1, A2, we see that 10 `KIP` -learned images, for both RBF and FC1, has comparable performance to several thousand natural images, thereby achieving a compression ratio of over 100. This compression ratio narrows as the support size increases towards the size of the training data.

Next, Table A3 compares FC1, RBF, and other kernels trained on all of MNIST to FC1 and RBF trained on `KIP` -learned images. We see that our `KIP` approach, even with 10K images (which fits into memory), leads to RBF and FC1 matching the performance of convolutional kernels on the original 60K images. Table A4 shows state of the art of FC kernels on CIFAR-10. The prior state of the art used kernel ensembling on batches of augmented data in Lee et al. (2020) to obtain test accuracy of 61.5% (32 ensembles each of size 45K images). By distilling augmented images using `KIP` , we are able to obtain 64.7% test accuracy using only 10K images.

## E.2   `KIP` AND `LS` TRANSFER ACROSS KERNELS

Figure A1 plots how `KIP` (with only images learned) performs across kernels. There are seven training scenarios: training individually on FC1, FC2, FC3, Conv1, Conv2, Conv3 NTK kernels and random sampling from among all six kernels uniformly (Avg All). Datasets of size 10, 100, 200

Table A2: **Accuracy on random subsets of CIFAR-10**. Standard deviations over 20 re-samplings.

| # Images \ Kernel | Linear | RBF | FC1 | CNTK[†] | Myrtle10-G[‡] |
|---|---|---|---|---|---|
| 10 | 16.2±1.3 | 15.7±2.1 | 16.4±1.8 | 15.33 ± 2.43 | 19.15 ± 1.94 |
| 20 | 17.1±1.6 | 17.1±1.7 | 18.0±1.9 | 18.79 ± 2.13 | 21.65 ± 2.97 |
| 40 | 17.8±1.6 | 19.7±1.8 | 20.6±1.8 | 21.34 ± 1.91 | 27.20 ± 1.90 |
| 80 | 18.6±1.5 | 23.0±1.5 | 23.9±1.6 | 25.48 ± 1.91 | 34.22 ± 1.08 |
| 160 | 18.5±1.4 | 25.8±1.4 | 26.5±1.4 | 30.48 ± 1.17 | 41.89 ± 1.34 |
| 320 | 18.1±1.1 | 29.2±1.2 | 29.9±1.1 | 36.57 ± 0.88 | 50.06 ± 1.06 |
| 640 | 16.8±0.8 | 32.8±0.9 | 33.4±0.8 | 42.63 ± 0.68 | 57.60 ± 0.48 |
| 1280 | 15.1±0.5 | 35.9±0.7 | 36.7±0.6 | 48.86 ± 0.68 | 64.40 ± 0.48 |
| 2560 | 13.0±0.5 | 39.1±0.7 | 40.2±0.7 | - | - |
| 5000 | 17.8±0.4 | 42.1±0.5 | 43.7±0.6 | - | - |
| 10000 | 24.9±0.6 | 45.3±0.6 | 47.7±0.6 | - | - |

[†] Conv14 kernel with global average pooling (Arora et al., 2019b)
[‡] Myrtle10-Gaussian kernel (Shankar et al., 2020)

Table A3: **Classification performance on MNIST.** Our `KIP`-datasets, fit to FC1 or RBF kernels, outperform non-convolutional kernels trained on all training images.

| Kernel | Method | Accuracy |
|---|---|---|
| FC1 | Base[1] | 98.6 |
| ArcCosine Kernel[2] | Base | 98.8 |
| Gaussian Kernel | Base | 98.8 |
| FC1 | KIP (a+l),[3] 10K images | 99.2 |
| LeNet-5 (LeCun et al., 1998) | Base | 99.2 |
| RBF | KIP (a+l), 10K images | 99.3 |
| Myrtle5 Kernel (Shankar et al., 2020) | Base | 99.5 |
| CKN (Mairal et al., 2014) | Base | 99.6 |

[1] Base refers to training on entire training dataset of natural images.
[2] Non RBF/FC numbers taken from (Shankar et al., 2020)
[3] (a + l) denotes `KIP` with augmentations and label learning during training.

are thereby trained then evaluated by averaging over all of FC1-3, Conv1-3, both with the NTK and NNGP kernels for good measure. Moreover, the FC and Conv train kernel widths (1024 and 128) were swapped at test time (FC width 128 and Conv width 1024), as an additional test of robustness. The average performance is recorded along the y-axis. AvgAll leads to overall boost in performance across kernels. Another observation is that Conv kernels alone tend to do a bit better, averaged over the kernels considered, than FC kernels alone.

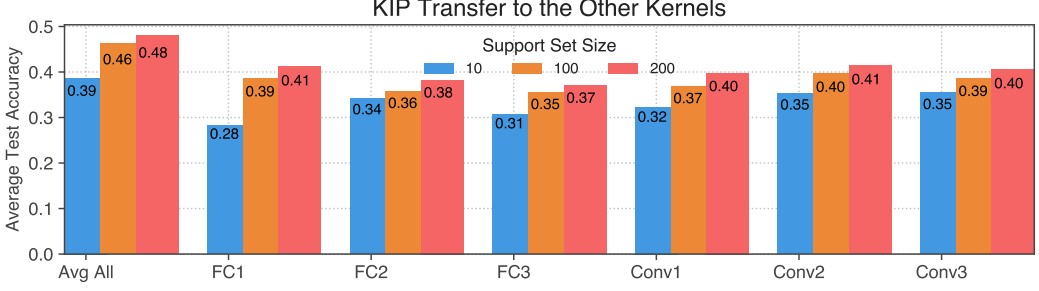

Figure A1: **Studying transfer between kernels.**

Table A4: **CIFAR-10 test accuracy for FC/RBF kernels.** Our `KIP`-datasets, fit to RBF/FC1, outperform baselines with many more images. Notation same as in Table A3.

| Kernel | Method | Accuracy |
|--------|--------|----------|
| FC1 | Base | 57.6 |
| FC3 | Ensembling (Lee et al., 2020) | 61.5 |
| FC1 | KIP (a+l), 10k images | **64.7** |
| RBF | Base | 52.7 |
| RBF | KIP (a+l), 10k images | **66.3** |

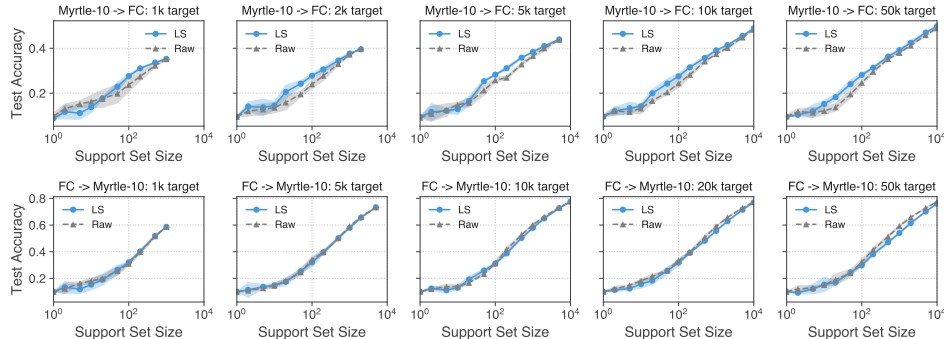

Figure A2: **Label Solve transfer between Myrtle-10 and FC for CIFAR10.** Top row: `LS` labels using Myrtle-10 applied to FC1. Bottom row: `LS` labels using FC1 applied to Myrtle-10. Results averaged over 3 samples per support set size. In all these plots, NNGP kernels were used and Myrtle-10 used regularized ZCA preprocessing.

In Figure A2, we plot how `LS` learned labels using Myrtle-10 kernel transfer to the FC1 kernel and vice versa. We vary the number of targets and support size. We find remarkable stability across all these dimensions in the sense that while the gains from `LS` may be kernel-specific, `LS`-labels do not perform meaningfully different from natural labels when switching the train and evaluation kernels.

### E.3 KIP TRANSFER TO NEURAL NETWORKS AND CORRUPTION EXPERIMENTS

Table A5: **KIP transfer to NN vs NN baselines on MNIST.** For each group of four experiments, the best number is marked boldface, while the second best number is in italics. Corruption refers to $90\%$ noise corruption. KIP images used FC1-3, Conv1-2 kernel during training.

| Method | 10 uncrpt | 10 crpt | 100 uncrpt | 100 crpt | 200 uncrpt | 200 crpt |
|--------|-----------|---------|------------|----------|------------|----------|
| FC1, KIP | **73.57±1.51** | *44.95±1.23* | **86.84±1.65** | *79.73±1.10* | **89.55±0.94** | *83.38±1.37* |
| FC1, Natural | 42.28±1.59 | 35.00±2.33 | 72.65±1.17 | 45.39±2.25 | 81.70±1.03 | 54.20±2.61 |
| LeNet, KIP | **59.69±8.98** | 38.25±6.42 | **87.85±1.46** | 69.45±3.99 | **91.08±1.65** | 70.52±4.39 |
| LeNet, Natural | *48.69±4.10* | 30.56±4.35 | *80.32±1.26* | 59.99±0.95 | *89.03±1.13* | 62.00±0.94 |

Table A6: **KIP transfer to NN vs NN baselines on CIFAR-10.** Notation same as in Table A5.

| Method | 100 uncrpt | 100 crpt |
|---|---|---|
| FC3, KIP | **43.09±0.20** | *37.71±0.38* |
| FC3, Natural | 24.48±0.15 | 18.92±0.61 |
| Conv2, KIP | **43.68±0.46** | *37.08±0.48* |
| Conv2, Natural | 26.23 ±0.69 | 17.10±1.33 |
| WideResNet, KIP | **33.29±1.14** | 23.89±1.30 |
| WideResNet, Natural | *27.93±0.75* | 19.00±1.01 |

Table A7: **MNIST. KIP and natural images on FC1. MSE Loss.** Test accuracy of image datasets of size 1K, 5K, 10K, trained using FC1 neural network using mean-square loss. Dataset size, noise corruption percent, and dataset type are varied: natural refers to natural images, KIP refers to `KIP`-learned images with either augmentations only (a) or both augmentations with label learning (a + l). Only FC1 kernel was used for `KIP`. For each KIP row, we place a * next to the most corrupt entry whose performance exceeds the corresponding 0% corrupt natural images. For each dataset size, we boldface the best performing entry.

| Dataset | 0% crpt | 50% crpt | 75% crpt | 90% crpt |
|---|---|---|---|---|
| Natural 1000 | 92.8±0.4 | 87.3±0.5 | 82.3±0.9 | 74.3±1.4 |
| KIP (a) 1000 | 94.5±0.4 | 95.9±0.1 | 94.4±0.2* | 92.0±0.3 |
| KIP (a+l) 1000 | **96.3±0.2** | 95.9±0.3 | 95.1±0.3 | 94.6±1.9* |
| Natural 5000 | 96.4±0.1 | 92.8±0.2 | 88.5±0.5 | 80.0±0.9 |
| KIP (a) 5000 | 97.0±0.6 | 97.1±0.6 | 96.3±0.2 | 96.6±0.4* |
| KIP (a+l) 5000 | **97.6±0.0*** | 95.8±0.0 | 94.5±0.4 | 91.4±2.3 |
| Natural 10000 | 97.3±0.1 | 93.9±0.1 | 90.2±0.1 | 81.3±1.0 |
| KIP (a) 10000 | 97.8±0.1* | 96.1±0.2 | 95.8±0.2 | 96.0±0.2 |
| KIP (a+l) 10000 | **97.9±0.1*** | 95.8±0.1 | 94.7±0.2 | 88.1±3.5 |

Table A8: **MNIST. KIP and natural images on FC1. Cross Entropy Loss.** Test accuracy of image datasets trained using FC1 neural network using cross entropy loss. Notation same as in Table A7.

| Dataset | 0% crpt | 50% crpt | 75% crpt | 90% crpt |
|---|---|---|---|---|
| Natural 1000 | 91.3±0.4 | 86.3±0.3 | 81.9±0.5 | 75.0±1.3 |
| KIP (a) 1000 | **95.9±0.1** | 95.0±0.1 | 93.5±0.3* | 90.9±0.3 |
| Natural 5000 | 95.8±0.1 | 91.9±0.2 | 87.3±0.3 | 80.4±0.5 |
| KIP (a) 5000 | **98.3±0.0** | 96.8±0.8* | 95.5±0.3 | 95.1±0.2 |
| Natural 10000 | 96.9±0.1 | 93.8±0.1 | 89.6±0.2 | 81.3±0.5 |
| KIP (a) 10000 | **98.8±0.0** | 97.0±0.0* | 95.2±0.2 | 94.7±0.3 |

Table A9: **CIFAR-10. KIP and natural images on FC1. MSE Loss.** Test accuracy of image datasets trained using FC1 neural network using mean-square loss. Notation same as in Table A7.

| Dataset | 0% crpt | 50% crpt | 75% crpt | 90% crpt |
|---|---|---|---|---|
| Natural 1000 | 34.1±0.5 | 34.3±0.4 | 31.7±0.5 | 27.7±0.8 |
| KIP (a) 1000 | **48.0±0.5** | 46.7±0.2 | 45.7±0.5 | 44.3±0.5* |
| KIP (a+l) 1000 | 47.5±0.3 | 46.7±0.8 | 44.3±0.4 | 41.6±0.5* |
| Natural 5000 | 41.4±0.6 | 41.3±0.4 | 37.2±0.2 | 32.5±0.7 |
| KIP (a) 5000 | **51.4±0.4** | 50.0±0.4 | 48.8±0.6 | 47.5±0.3* |
| KIP (a+l) 5000 | 50.6±0.5 | 48.5±0.9 | 44.7±0.6 | 43.4±0.5* |
| Natural 10000 | 44.5±0.3 | 43.2±0.2 | 39.5±0.2 | 34.3±0.2 |
| KIP (a) 10000 | **53.3±0.8** | 50.5±1.3 | 49.4±0.2 | 48.2±0.6* |
| KIP (a+l) 10000 | 51.9±0.4 | 50.0±0.5 | 46.5±1.0 | 43.8±1.3* |

Table A10: **CIFAR-10. KIP and natural images on FC1. Cross Entropy Loss.** Test accuracy of image datasets trained using FC1 neural network using cross entropy loss. Notation same as in Table A7.

| Dataset | 0% crpt | 50% crpt | 75% crpt | 90% crpt |
|---|---|---|---|---|
| Natural 1000 | 35.4±0.3 | 35.4±0.3 | 31.7±0.9 | 27.2±0.8 |
| KIP (a) 1000 | **49.2±0.8** | 47.6±0.4 | 47.4±0.4 | 45.0±0.3* |
| Natural 5000 | 43.1±0.8 | 42.0±0.2 | 38.0±0.4 | 31.7±0.6 |
| KIP (a) 5000 | 44.5±1.0 | **51.5±0.3** | 51.0±0.4 | 48.9±0.4* |
| Natural 10000 | 45.3±0.2 | 44.8±0.1 | 40.6±0.3 | 33.8±0.2 |
| KIP (a) 10000 | 46.9±0.4 | **54.0±0.3** | 52.1±0.3 | 49.9±0.2* |

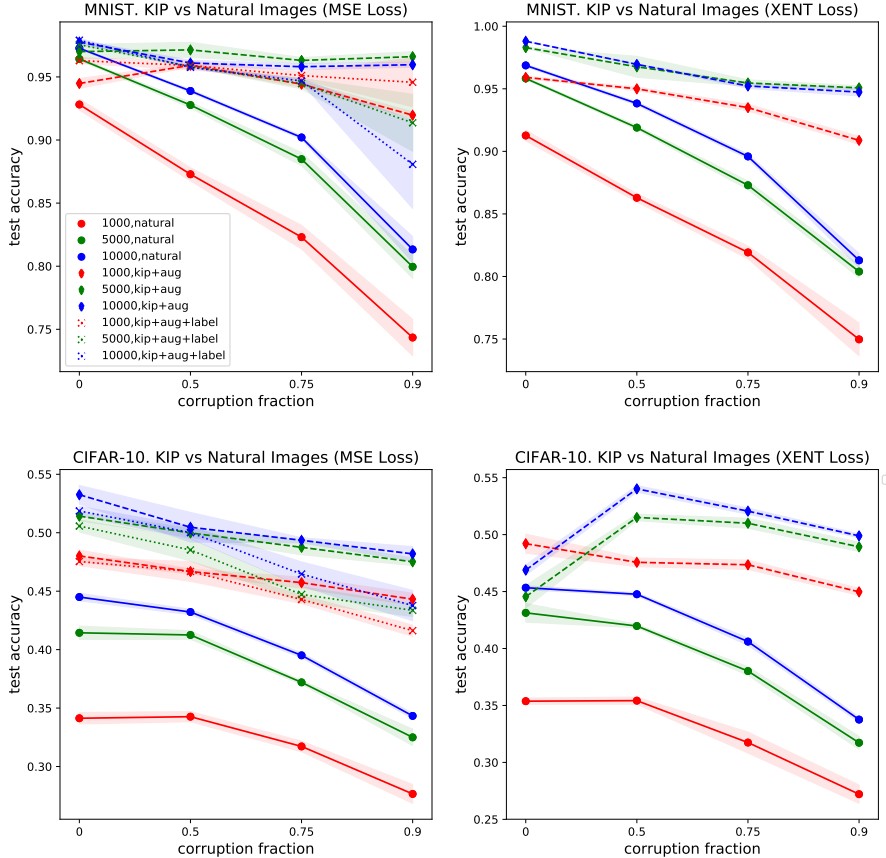

Figure A3: **KIP vs natural images, FC1.** Data plotted from Tables A7-A10, showing natural images vs. `KIP` images for FC1 neural networks across dataset size, corruption type, dataset type, and loss type. For instance, the upper right figure shows that on MNIST using cross entropy loss, 1k `KIP` + aug learned images with 90% corruption achieves 90.9% test accuracy, comparable to 1k natural images (acc: 91.3%) and far exceeding 1k natural images with 90% corruption (acc: 75.0%). Similarly, the lower right figure shows on CIFAR10 using cross entropy loss, 10k `KIP` + aug learned images with 90% corruption achieves 49.9%, exceeding 10k natural images (acc: 45.3%) and 10k natural images with 90% corruption (acc: 33.8%).

## F  EXAMPLES OF KIP LEARNED SAMPLES

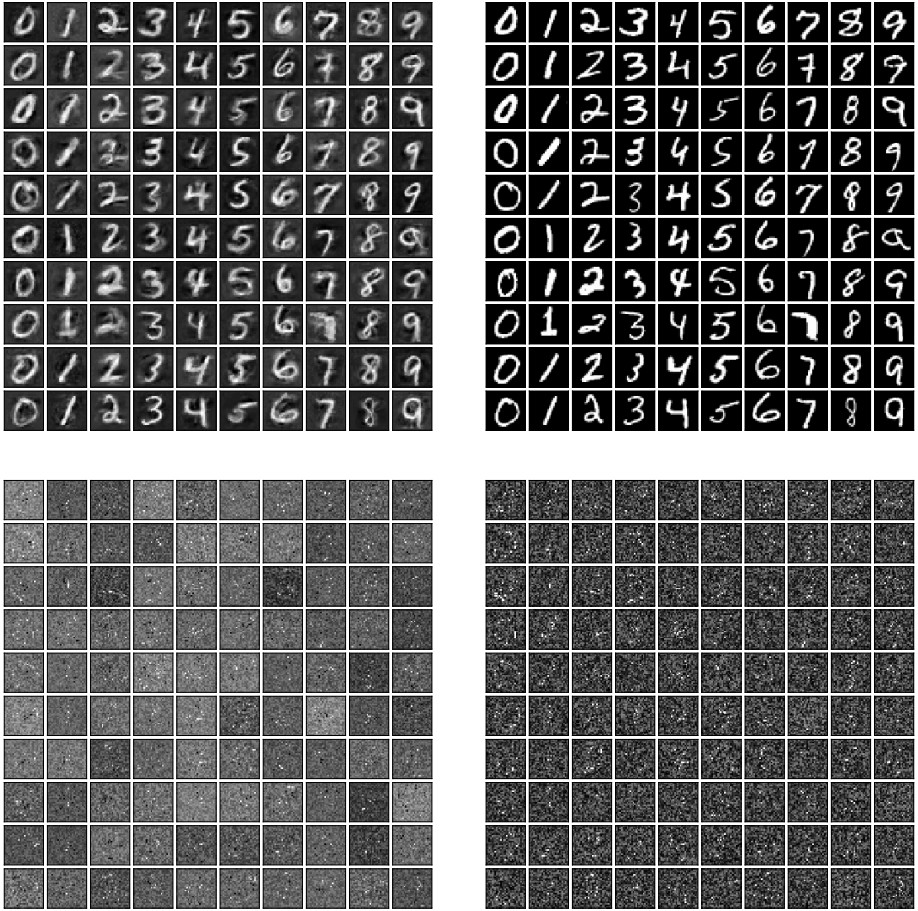

Figure A4: **KIP learned images (left) vs natural MNIST images (right).** Samples from 100 learned images. Top row: 0% corruption. Bottom row: 90% noise corruption.

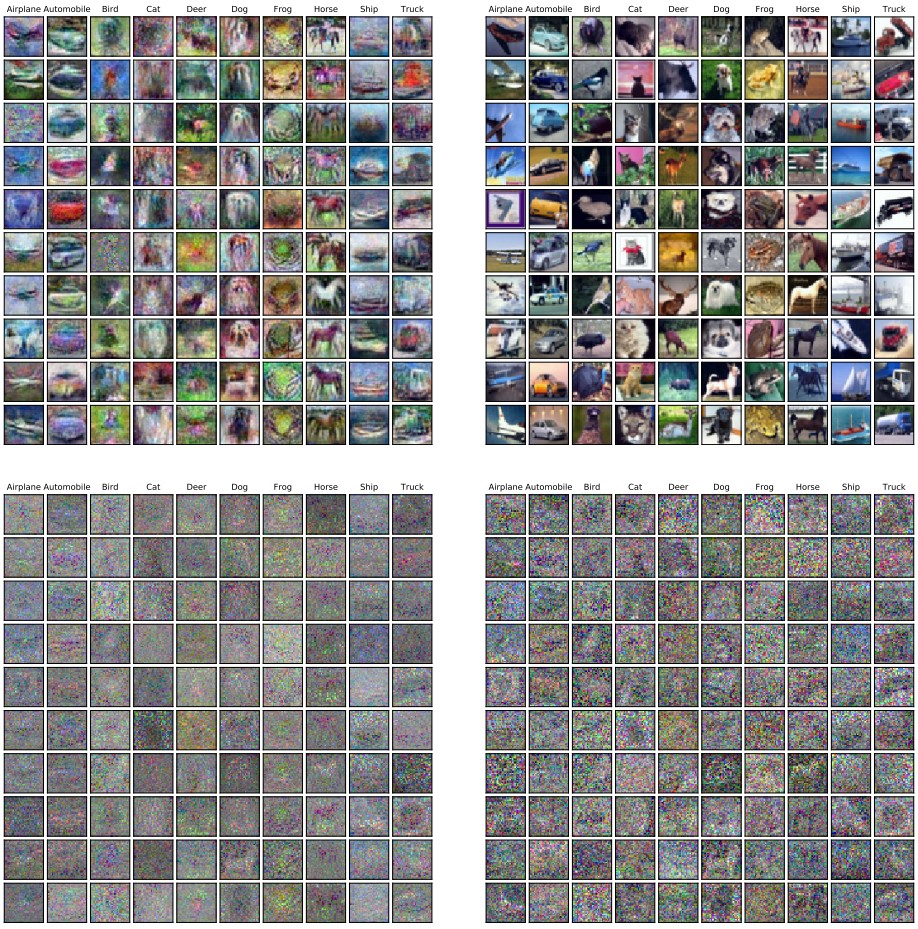

Figure A5: **KIP learned images (left) vs natural CIFAR-10 images (right).** Samples from 100 learned images. Top row: 0% corruption. Bottom row: 90% noise corruption.

