# OpenReview forum: "Dataset Meta-Learning from Kernel Ridge-Regression"
_ICLR.cc/2021/Conference — ICLR 2021 Poster_

### Official Review · AnonReviewer1 · 2020-10-28
**Interesting yet not efficient**

**Rating:** 4
**Confidence:** 4

**Review:**

The paper proposed an objective based on kernel ridge regression to find a coreset of the training set. The objective first adopts gradient descent to find the coreset in the input space, then generates targets for fake samples in the coreset. I do have many comments on the claims made in this paper, and I hope the authors can answer them.

1. If the goal is to find a minimum number of samples, either original or fake ones, why is the definition of \epsilon-approx measuring the difference between two expected losses under two distributions respectively, not the difference between two conditional distributions y|x or the joint distributions x,y ? Also, the loss function defined in Eq. 4 clearly minimises the distance between two conditional distributions.

One might be able to find a coreset of samples that in expectation gives a lower loss value than the original dataset, e.g. denoising, so then measuring the difference between loss values doesn't seem reasonable to me.

2. From a CS theory perspective, the \epsilon-kernel coreset has a clear definition, and please refer to [1] for details.

3. The two examples given are not precise.

For example 1, the described situation only holds when the original dataset is linearly separable. When the dataset is not linearly separable, once SVM determines the support vectors on the original dataset, there are samples in between two margins, therefore, a new SVM which is only trained on the samples with support vectors and samples in between will not give you 0-approx by the definition in the paper.

For example 2, from a statistical learning perspective, the most important part for linear/ridge regression models is the covariance structure derived from X, so then it is true that one can always find a 0-approx dataset, but not of any size. In terms of privacy related issues, regression models rarely have privacy issues as only the X^T X and X^T Y are kept, the information regarding to specific data samples is eliminated due to the inner product. Therefore, the claim that the proposed algorithm can help improve privacy isn't valid,  as linear/ridge regression models already do that.

4. There are many ways of computing inducing points of a given dataset without learning, including random projections[2], frequent directions[3,4], other subsampling methods for Nystroem[5], etc. Many of those methods are well-studied with nice theoretical guarantee in reconstruction. The paper largely ignored those methods in comparison.

5. I don't see how NTK is necessary here. If the authors are interested in testing whether their proposed method work, they can simply start from simulated data in 2D space, and illustrate that the proposed method is indeed able to pick out a high-quality coreset than other methods. Then one can quickly run experiments using pretrained neural networks as feature extractors, as was done in [6]. Given different feature extractors, the selected coresets from individual one of them give us a probe to understand how different neural networks are learnt.

[1] Phillips, J. M. (2016). Coresets and sketches. arXiv preprint arXiv:1601.00617.
[2] Woodruff, D. P. (2014). Sketching as a tool for numerical linear algebra. arXiv preprint arXiv:1411.4357.
[3] Ghashami, M., Liberty, E., Phillips, J. M., & Woodruff, D. P. (2016). Frequent directions: Simple and deterministic matrix sketching. SIAM Journal on Computing, 45(5), 1762-1792.
[4] Shi, B., & Phillips, J. M. (2020). A Deterministic Streaming Sketch for Ridge Regression. arXiv preprint arXiv:2002.02013.
[5] Kumar, S., Mohri, M., & Talwalkar, A. (2012). Sampling methods for the Nyström method. The Journal of Machine Learning Research, 13(1), 981-1006.
[6] Tang, S., & de Sa, V. R. (2020). Deep Transfer Learning with Ridge Regression. arXiv preprint arXiv:2006.06791.

---

> ### Author Response · Authors · 2020-11-13
> **Response to Reviewer #1 (part 1)**
>
> We thank the reviewer for their time and all the questions and references provided. There appears to be some misinterpretation of the paper’s content which we hope to clarify:
>
> 1. We are unsure why the reviewer claims we work with conditional distributions (Note: Some typos in Def. 2 that have been corrected). We compare loss with respect to the joint distribution (x,y) of input-label pairs. For comparison, in classification problems using neural networks (NNs), a conditional distribution p(y|x) is predicted for each x in training, and it is optimized with respect to cross entropy loss. But for test eval, samples from the joint distribution of input-labels are used to evaluate performance. Our use of the loss function (for both eps-approximation and KIP) is as in this latter situation: we sample from a joint distribution. In practice, we end up replacing population data with a fixed target dataset, i.e., the empirical distribution.
> 2. Regarding expectation of loss values vs comparing loss values directly, this is addressed by our notion of weak vs strong eps-approximation. As after Def. 2, these notions converge for 0-1 loss for epsilon small. Please clarify if this does not address the reviewer's concerns.
> 3. Reference [1] was helpful in alerting the authors to prior work. Note however that the weak corset def. of [1] applies to unsupervised clustering whereas we consider supervised learning. [1] is now cited in Relevant Work and Appendix A.
>
>    We thank the reviewer for catching the imprecision of Ex. 1 and 2. Indeed, our SVM example assumed linear separability (we have removed this assumption in the current version). However, we are confused by the reviewer’s remark that in the non-separable case, retraining an SVM on the support vectors and the data between the margins (or more precisely, those with positive slack) does not yield an equivalent SVM (what 0-approximate dataset entails). Perhaps the reviewer misunderstood our definition of 0-approximation.
>
>    For the case of ridge-regression, we were imprecise in that we implicitly assumed the scalar output case. Our revision discusses the general case with a detailed proof: for k-classes, an 0-approximate dataset of size k exists.
>
>    Finally, the remark about privacy not being an issue for linear models, while true, does not detract from our work. Linear models are very limited in terms of their expressivity: the real challenge is to find an expressive model that preserves privacy. Neural networks are highly expressive, but their privacy guarantees are unclear and the subject of ongoing research. For some ways in which the data used to train a network can be approximately recovered, see https://blog.openmined.org/extracting-private-data-from-a-neural-network. By training on corrupt data as we proposed, we can hope to nullify such attacks. Secondly, kernel methods still require access to the training data in order to construct test-train kernel elements for making predictions, so storing train-train kernel products isn't sufficient.
> 4. The references [2-5] provided, while relevant for approximation, both 1) do not create datasets; 2) have no obvious application to privacy-preservation, and thus lie outside the novelty of our paper. For 1), one of our main results was to show that our learned datasets transfer to other kernels or neural networks. Doing kernel matrix approximation via [2-5], no dataset is obtained to be used elsewhere. For 2), it is not clear how [2-5] could be used with privacy-enhancing corruption. Part of our novelty is that even with high corruption rates of 90%, we could learn data that outperforms clean data. Kernel matrix approx. methods, whose performance cannot exceed the original matrix, will do poorly on corrupt data. We summarize this point and include it in the Related Work.
> 5. While KIP and LS are pure kernel methods with no-reference to neural networks (NN), we strongly believe transferability to deep NNs is one of the key interesting findings with potential wide applications. The connection between the infinite-width limit of neural networks to NTK is the grounding theory that allows this bridge between kernels and NNs: thus KIP leads to much more sample-efficient training of NNs from scratch.
>
>    Regarding the proposed simple cases for toy 2d data, while they can be interesting to study, we believe the power of KIP lies in learning high-dimensional data effectively, thus our focus on high-dimensional image inputs. Moreover, we are having a hard time connecting [6] to our study since [6] mostly uses pre-trained networks features for KRR. To our understanding, the reviewer's proposed method is a very indirect way of testing whether KIP learned data inputs are useful for NNs. We hope the reviewer could clarify further, if we are missing the point.
>
> Finally, please see part 2 for an overview of the what's been added in the revision.

---

> ### Author Response · Authors · 2020-11-13
> **Response to Reviewer #1 (part 2)**
>
> Please note our revised paper which, in addition to addressing reviewer concerns
> * corrects typos
> * improves exposition (including the description of tables/figures and the Related Work section, which has been moved towards the end)
> * contains additional experiments concerning corrupt vs natural images showing that even for larger dataset sizes, 90% corrupted KIP images continue to outperform natural images (Tables A7-A10, Figure A3); some mixed results are also presented for labels learned using KIP
> * adds the analogue of Theorem 1 for Label Solve (Theorem 2), relating the latter to epsilon-approximation

---

### Official Review · AnonReviewer2 · 2020-10-28
**Paper present a novel approach for approximate compression of datasets using Kernel Ridge Regression and experimental results show efficacy of the approach in terms of reducing sample complexity.**

**Rating:** 7
**Confidence:** 4

**Review:**

Paper present a novel approach for approximate compression of datasets using Kernel Ridge Regression, referred to as KIP.  Paper is well written and technically sounds and experimental results show efficacy of the approach in terms of reducing sample complexity. It also provides an added benefit of corrupting input datasets without much loss test accuracy for privacy preserving use case learning.

The definition of $\epsilon$-approximation is introduced in terms bounds on difference between the expected empirical loss for original and compressed dataset which in potentially also bounds generalization error as well. The KIP algorithm iterates over an initial support-set to finally converge over a support dataset that gives similar test performance. The idea of choosing a support set and growing sounds familiar to Nystorm method for kernel approximation provides intuition on  why the approach might work.  Also selection of multiple base kernels also means selection of multiple feature spaces which naturally leads to overall boost in performance. Results on MNIST and CFIR shows the efficacy of these results.

A potential gap in this work is the trade-off between the compressed dataset size N and  test error for a given fixed $\epsilon$. Is there a way to choose say N, for a given $\epsilon$ for a given test error performance.  It may be so that we end choosing all points in the original dataset for a given approximation and test error. A characterization of this using generalization bounds would be helpful.

---

> ### Author Response · Authors · 2020-11-13
> **Response to Reviewer #2**
>
> We thank the reviewer for their time and constructive feedback on the submission. Regarding the suggestion about generalization bounds: this would make for an excellent direction for followup work and is perhaps out of scope for the current paper. The notion of epsilon-approximation is specific to the pair of algorithms used to compare the original and approximating dataset. As our SVM and ridge regression examples after Definition 2 show (please see revised version for details), the relationship between $N$ and $\epsilon$ can have markedly different behaviors (for SVM, one can ensure $O(N)$ for $\epsilon = 0$ while for ridge regression, one can get $O(1)$ for $\epsilon = 0$). Of course, the interesting case to study is for neural networks. However, given the difficulty of ascertaining meaningful generalization bounds for neural networks, we do not have something concrete and useful to say about the subject at this time.
>
> Please note our revised paper which, in addition to addressing reviewer concerns
> * corrects typos
> * improves exposition (including the description of tables/figures and the Related Work section, which has been moved towards the end)
> * clarifies the examples of epsilon-approximation after Definition 2
> * contains additional experiments concerning corrupt vs natural images showing that even for larger dataset sizes, 90% corrupted KIP images continue to outperform natural images (Tables A7-A10, Figure A3); some mixed results are also presented for labels learned using KIP
> * adds the analogue of Theorem 1 for Label Solve (Theorem 2), relating the latter to epsilon-approximation

---

### Official Review · AnonReviewer4 · 2020-10-28
**Recommendation to Accept**

**Rating:** 6
**Confidence:** 3

**Review:**

This paper proposes a data-driven approach to choose an informative surrogate sub-dataset,  termed "a \epsilon-approximation", from the original data set. A meta-learning algorithm called Kernel Inducing Points (KIP ) is proposed to obtain such sub-datasets for (Linear) Kernel Ridge Regression (KRR), with the potential to extend to other machine learning algorithms such as neural networks.  Some theoretical results are provided for the KRR with a linear kernel. The empirical performance of the proposed algorithm is evaluated by experiments based on synthetic data and some standard benchmark data sets.

Overall, I think the paper is well written and the proposed method is of potential value to existing literature.

Strengths:
1. Some theoretical results are provided for the KIP algorithm under the linear KRR setting.
2. Numerical experiments are carefully designed and appear to be convincing.

Weakness:
1. The theoretical results are only for Linear KRR. However, this model cannot be used for classification with categorical labels (when the response y is categorical variables). When the classification is involved
2. The KIP algorithm depends on (1) the choice of the kernel; (2) the choice of Loss function; (3) the choice of tuning parameters for the kernel machine.  If we choose the sub-dataset based on one set of rules and later want to use it for some other purposes, this certainly would cause some problems. How can you address this issue? In other words, how the proposed algorithm can be adapted to obtain a sub-dataset that works for multi-purposes?
3. Step 5 in Algorithm 1 is not very clear to me. How exactly do you decide to add a batch of data into the support set? based on what threshold/criterion? If one has to evaluate the loss function (4) on all possible subset of size N out of a total sample size n, it will cost about {n \choose N} number of operations, which is clearly not scalable when n is large.  More details on how to update the support set and the convergence criterion.

Other issues:
1. There are typos in multiple places such as equations (1) and (3), and in Definition 3. Please proofread carefully.
2. For the classification problems, it appears that the support vector machine is always the best choice seems it is "0-approximation" to the original dataset. Please clarify why and why not.

---

> ### Author Response · Authors · 2020-11-13
> **Response to Reviewer #4**
>
> We thank the reviewer for their time and feedback. Concerning the points raised in the Weaknesses section:
>
> 1. All the problems we consider in our paper are classification problems involving one-hot labels. For KRR (linear or otherwise), we use mean-centered versions (e.g. for 10-way classification, a vector with 0.9 on the target class and  -0.1 on the remaining nine). We have clarified in the current revision that regression can be turned into prediction by predicting the class with the largest value (second paragraph of Section 4). While our Theorem 1 only holds for Linear KRR, we think it is a necessary and encouraging validating step for the soundness of KIP (in the same way that having a sound theory for linear classifiers is the starting point for the much less tractable case of non-linear neural networks.) Note that even for Linear KRR, the objective is highly nonlinear in the support set and is not a trivial result.
> 2. Thanks for raising a great question. It has been observed in the few-shot literature that training with multiple support sizes works well, and is probably helpful for robustness at evaluation time with regards to the number of support data used (i.e. varying the n-shot, k-way tasks). One can vary the support size used during KIP, but we focused on the case of sampling from the entire support set in order to get optimal compression results at predetermined dataset sizes. Thus, to get robustness to sub-dataset selection, random support sampling (Step 5 of Algorithm 1) with small, medium, and large dataset sizes can be used.
> 3. Only the randomly selected subset of the support (bar X, bar y) is updated per step in the while loop; the non-sampled remaining images in the support set remain fixed at such a step. One can think of the entire support set as the set of parameters we are learning. At each step, we update some random subset of those parameters via an arbitrary random sampling scheme (in the paper, we end up sampling all the parameters). We have updated the text to make this point more clear.
>
> For the other issues:
>
> 1. We’ve updated the submission to correct typos to the best of our ability. We hope the new version improves clarity.
> 2. Could the reviewer please further clarify the comment on SVM. If by “best” the reviewer means that one does not need to add any additional datapoints beyond the support vectors (in the linearly separable case) to obtain 0-approximation, then this is indeed true. (Note: we have updated Example 1 on SVM for the general case of data that is not linearly separable)
>
> Please note our revised paper which, in addition to addressing reviewer concerns
>
> * improves exposition (including the description of tables/figures and the Related Work section, which has been moved towards the end)
> * clarifies the examples of epsilon-approximation after Definition 2
> * contains additional experiments concerning corrupt vs natural images showing that even for larger dataset sizes, 90% corrupted KIP images continue to outperform natural images (Tables A7-A10, Figure A3); some mixed results are also presented for labels learned using KIP
> * adds the analogue of Theorem 1 for Label Solve (Theorem 2), relating the latter to epsilon-approximation

---

### Official Review · AnonReviewer3 · 2020-10-29
**Official Blind Review #3**

**Rating:** 6
**Confidence:** 3

**Review:**

This paper tackles the problem of data compression using kernel induced points learned through deep kernel learning. The main contributions of the paper are :

1 - the formalism of the notion of $\epsilon$ - approximation of datasets that describes how to rely on one learning algorithm (here KRR) to find the compression set for another one, while keeping the generalization error below $\epsilon$.

2 - the proposal of the KPI and LS algorithms that find the pseudo-inputs set by leveraging kernel methods and by modifying respectively the inputs and the labels of those pseudo-inputs while monitoring the loss of a regressor using them on the original dataset that is being compressed. Since the starting pseudo-inputs can be corrupted versions of the original dataset, the authors advocate that their method preserves privacy.

On Quality: The paper is well written and structured though could benefit from more details at many places (e.g. most figures and table  descriptions are lacking or not detailed enough)

On Clarity: The intention and the contribution of the authors are clear. The main concerning point is linking their work to too many areas instead of focusing on the main scope of the paper which is explicit data compression/distillation. By doing so, the authors confuse the reader especially given that the related work is just after the intro. Another confusing aspect is the usage of terms like meta-learning to describe their algorithm when the algorithm only deals with a single dataset (episodic-looping through one dataset is not meta-learning). Clearer language  and simplification of related work could go along way

On originality: the epsilon-approximation is an interesting idea that could lead to interesting algorithm developments. I believe the idea is not novel but the authors are the first to formalize it. On the usage of deep kernel learning to do data distillation, the idea has been explored too (Snelson & Ghahramani, 2006) but the authors took it a bit further with the LS algorithm and the privacy preservation aspect.

On significance: The area is an important research area and there are definitely a few ideas here that others can build upon.

Technical concerns/questions:
- In theorem 1: how do you enforce Eq A1? do you make sure that the norm of pseudo-inputs are less than 1 ?
- in def3: do you need D*_N to define the compression ratio? why not define it using D?
- In the label solving, after a batch the label solved can fall outside of the possible label set. How do you handle that? or it does not matter?

---

> ### Author Response · Authors · 2020-11-13
> **Response to Reviewer #3**
>
> We thank the reviewer for their time and constructive feedback to improve the paper.
>
> Our response to the technical questions:
> 1. The norm condition in Theorem 1 on the input is from the distribution P for which we evaluate loss (and is merely a convenience for bounding the value of the mean squared error for linear models). It is not a condition for the eps-approximate dataset to be learned: the resulting dataset can have data of arbitrary norm.
> 2. We define the compression ratio using subsets of D because D might be highly redundant/simple. (Note our definition of compression ratio has been updated in the revision to be a heuristic definition.) For example, if a 100x smaller subset of D performs nearly as well as D, we might not want to include that factor of 100 in computing the compression ratio (a trivial example is if one has a small linear model trained on linearly dependent data).
> 3. We place no constraints on the labels learned by Label Solve. Since we use the labels for regression tasks, these labels are still valid for use. In fact, our current revision performs some experiments concerning the case of labels learned during KIP (alongside the images), please see comments at end. It is an interesting direction for future work to add a regularizer or additional constraints when labels are learned during KIP/solved using Label Solve.
>
> Concerning our use of the term meta-learning, we regard our method as meta-learning due to our bi-level optimization (Section 3 of the current version) where the inner optimization is by the KRR solver and the outer optimization is learning the support set fed into the KRR solver. Each update to the support set is thus one meta-update episode. While we agree that conventional meta-learning involves learning over multiple datasets, we believe the term is used more broadly nowadays and covers our setup, see e.g. [1].
>
> Please note our revised paper which, in addition to addressing reviewer concerns:
> * corrects typos
> * improves exposition (including the description of tables/figures and the Related Work section, which has been moved towards the end)
> * clarifies the examples of epsilon-approximation after Definition 2
> * contains additional experiments concerning corrupt vs natural images showing that even for larger dataset sizes, 90% corrupted KIP images continue to outperform natural images (Tables A7-A10, Figure A3); some mixed results are also presented for labels learned using KIP
> * adds the analogue of Theorem 1 for Label Solve (Theorem 2), relating the latter to epsilon-approximation.
>
> [1] Meta-Learning with Implicit Gradients (https://arxiv.org/abs/1909.04630)

---

### Author Response · Authors · 2020-11-21
**Changes in latest revision.**

In this third version, we
* clean up definitions in Section 2; most noticeably, we refactor the definition of epsilon-approximation into a first part relating function approximation and then a second part for dataset approximation
* revise our presentation of Label Solve (end of section 3)
* slight correction of Theorem 2 and slight cleanup of statements of Theorems 1 and 2

---

### Decision · Program_Chairs · 2021-01-07
**Final Decision**

**Decision:**

Accept (Poster)

**Comment:**

Three reviewers agree on the value of the contribution and recommend acceptance. A reviewer votes for rejection but the authors have clarified all the major concerns raised by the reviewer. Therefore, I recommend acceptance.